# Does Unsupervised Domain Adaptation Improve the Robustness of Amortized Bayesian Inference? A Systematic Evaluation

**Lasse Elsemüller**                                            *lasse.elsemueller@gmail.com*
*Heidelberg University, Germany*

**Valentin Pratz**
*Heidelberg University, Germany*
*Zuse School ELIZA, Germany*

**Mischa von Krause**
*Heidelberg University, Germany*

**Andreas Voss**
*Heidelberg University, Germany*

**Paul-Christian Bürkner**
*TU Dortmund University, Germany*

**Stefan T. Radev**                                            *stefan.radev93@gmail.com*
*Rensselaer Polytechnic Institute, USA*

**Reviewed on OpenReview:** *https://openreview.net/forum?id=ewgLuvnEw6*

## Abstract

Neural networks are fragile when confronted with data that significantly deviates from their training distribution. This is true in particular for simulation-based inference methods, such as neural amortized Bayesian inference (ABI), where models trained on simulated data are deployed on noisy real-world observations. Recent robust approaches employ unsupervised domain adaptation (UDA) to match the embedding spaces of simulated and observed data. However, the lack of comprehensive evaluations across different domain mismatches raises concerns about the reliability in high-stakes applications. We address this gap by systematically testing UDA approaches across a wide range of misspecification scenarios in silico and practice. We demonstrate that aligning summary spaces between domains effectively mitigates the impact of unmodeled phenomena or noise. However, the same alignment mechanism can lead to failures under prior misspecifications—a critical finding with practical consequences. Our results underscore the need for careful consideration of misspecification types when using UDA to increase the robustness of ABI.

## 1 Introduction

Synthetic data can augment numerous real-world applications (Savage, 2023), including complex statistical workflows (Bürkner et al., 2025). In line with this perspective, amortized Bayesian inference (ABI; Gershman & Goodman, 2014) redefines the classical sampling problem in Bayesian estimation by training generative neural networks on simulations derived from computational models (Bürkner et al., 2023; Cranmer et al., 2020). The trained neural networks are then deployed to efficiently solve inference tasks as diverse as inferring evolutionary parameters (Avecilla et al., 2022) or gravitational waves (Pacilio et al., 2024).

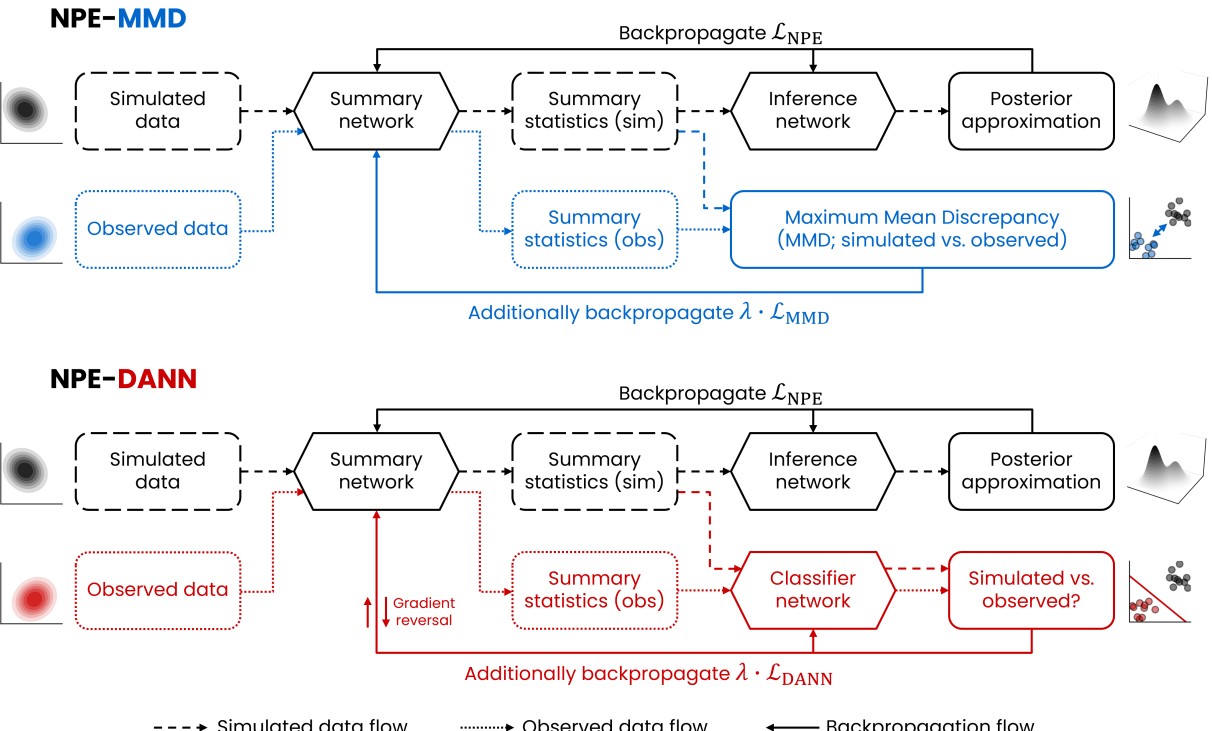

Figure 1: Schematic overview of NPE-UDA methods that combine neural posterior estimation (NPE) with unsupervised domain adaptation (UDA). Standard NPE training optimizes posterior approximation in a simulation-based training loop. NPE-UDA approaches introduce observed data into the training procedure, targeting performance improvements in the (possibly shifted) observed domain via *domain alignment in summary space*. NPE-MMD (maximum mean discrepancy) directly minimizes the distance between distributions, whereas NPE-DANN (domain-adversarial neural networks) uses adversarial competition between an auxiliary domain classifier and the summary network.

Evidently, the faithfulness of any simulation-based method rests on the critical assumption that statistical patterns learned from simulated data can be extrapolated to real observations. This assumption inevitably situates ABI in a domain-shift regime, exacerbated by the degree of potential mismatch between model simulations and reality. As such, *robustness to model misspecification* has been identified as the primary challenge for amortized methods in different fields (Dingeldein et al., 2024; Rainforth et al., 2024; Cannon et al., 2022).

Unsupervised domain adaptation (UDA) studies the transfer of knowledge from a labeled source domain to an unlabeled target domain. It aims to mitigate domain shifts by aligning the *embedding spaces* of the two domains. This property makes UDA a promising approach for addressing domain shifts in ABI, as the latter typically combines inference with embedding high-dimensional data into *learned summary statistics* (Radev et al., 2020; Chan et al., 2018). Indeed, recent research has underscored the critical role of *in-distribution* summary statistics for achieving robust simulation-based inference (Schmitt et al., 2024; Frazier et al., 2024; Huang et al., 2023; Wehenkel et al., 2024).

So far, only two pioneering studies (Swierc et al., 2024; Huang et al., 2023) have explored the potential of UDA methods for robustifying simulation-based inference. Both approaches align the embedding spaces by minimizing the maximum mean discrepancy (MMD; Gretton et al., 2012) between simulated and observed summary statistics (see Figure 1). However, despite their promising results, several gaps remain. In particular, Huang et al. (2023) did not make an explicit connection to UDA and explored a non-amortized approach. While Swierc et al. (2024) acknowledged the connection to UDA, their work focused on a specific gravita-

tional lensing application. Both works mainly evaluated likelihood misspecification, leaving the behavior under prior shifts largely untapped. Finally, the utility of the widely used UDA method *domain-adversarial neural networks* (DANN; Ganin et al., 2016) remains completely unexplored. To address these gaps, we make the following contributions:

1. We adapt domain-adversarial neural networks for neural posterior estimation (NPE; see Figure 1) and evaluate their utility for robust amortized Bayesian inference.

2. We theoretically categorize robust methods by inference targets, explaining when and why UDA-based ABI methods succeed or fail under different sources of model misspecification.

3. We empirically evaluate the robustness of UDA-based ABI methods across both prior and likelihood misspecification in multiple benchmarks, confirming the central role of the source of misspecification: whereas UDA improves performance under likelihood shifts, it is detrimental under prior shifts.

## 2 Background

**Amortized Bayesian Inference (ABI)**  Amortized methods (Gershman & Goodman, 2014; Ritchie et al., 2016; Le et al., 2017) are a subset of the simulation-based inference (SBI; Cranmer et al., 2020) family. Their defining characteristic is the ability to perform zero-shot inference on model parameters $\boldsymbol{\theta}$ by learning a conditional distribution $q(\boldsymbol{\theta} \mid \boldsymbol{x})$ that requires no further training or auxiliary algorithms for new data $\boldsymbol{x}$ (see Appendix A for details). The *amortized distribution* $q(\boldsymbol{\theta} \mid \boldsymbol{x})$ is typically parameterized by an inference network, a generative neural network that can generate random samples $\boldsymbol{\theta} \sim q(\boldsymbol{\theta} \mid \boldsymbol{x})$—akin to a standard Markov chain Monte Carlo (MCMC) sampler, but orders of magnitude faster. Often, the inference network is preceded by a summary network $\phi$ that compresses raw observations to learned summary statistics $\phi(\boldsymbol{x})$, leveraging probabilistic symmetries in the data (Radev et al., 2020; Chen et al., 2021). Following a potentially expensive simulation-based training phase, the network can be queried with any *new* data $\boldsymbol{x}_{\text{new}}$ to rapidly approximate the target distribution $p(\boldsymbol{\theta} \mid \boldsymbol{x}_{\text{new}})$. Initially dismissed as inefficient compared to sequential methods optimized for a specific data set $\boldsymbol{x}_{\text{obs}}$ (Papamakarios & Murray, 2016), amortized methods have since achieved notable successes across various domains (Bürkner et al., 2023; Zammit-Mangion et al., 2024).

**Unsupervised Domain Adaptation (UDA)**  UDA is a subfield of transductive transfer learning where labeled data is only available for the source domain $\mathcal{D}_S = \{(\boldsymbol{x}_S^i, \boldsymbol{y}_S^i)\}_{i=1}^{N_S}$, distributed according to $p_S(\boldsymbol{x}, \boldsymbol{y})$, but not for the target domain $\mathcal{D}_T = \{\boldsymbol{x}_T^i\}_{i=1}^{N_T}$, distributed according to $p_T(\boldsymbol{x}_T, \boldsymbol{y}_T)$ (Johansson et al., 2019). UDA methods are based on the seminal theoretical works of Ben-David et al. (2006; 2010), who introduced generalization bounds for binary classification tasks that bound the risk in the target domain $R_T$ of a hypothesis $h \in \mathcal{H}$:

$$R_T(h) \leq R_S(h) + d_{\mathcal{H}\Delta\mathcal{H}}(p_S, p_T) + \lambda_{\mathcal{H}}, \tag{1}$$

where $R_S(h)$ is the source domain risk, $d_{\mathcal{H}\Delta\mathcal{H}}(p_S, p_T)$ measures the divergence between the domain distributions, and $\lambda_{\mathcal{H}}$ is the minimum combined risk of the optimal hypothesis, $\lambda_{\mathcal{H}} = \inf_{h \in \mathcal{H}} |R_S(h) + R_T(h)|$ (Johansson et al., 2019). This suggests that domain adaptation from $\mathcal{D}_S$ to $\mathcal{D}_T$ can be facilitated by minimizing the divergence between the marginal domain distributions. Although the domain distribution divergence cannot be reduced directly, the representation divergence $d(\phi(\boldsymbol{x}_S), \phi(\boldsymbol{x}_T))$ from a transformation $\phi : \mathcal{X} \to \mathcal{Z}$ can be readily minimized (Ben-David et al., 2006). The core idea of UDA is thus twofold: (i) to minimize the source-domain error $R_S(h)$ during training, and (ii) to align the domain representations $\phi(\boldsymbol{x}_S)$ and $\phi(\boldsymbol{x}_T)$ to achieve *domain-invariant* embeddings that generalize to the target domain. UDA methods include discrepancy-based approaches, which minimize statistical divergences like the MMD between source and target embeddings (Tzeng et al., 2014), and, most prominently, adversarial-based approaches, such as domain-adversarial neural networks (DANN) (Ganin et al., 2016), which learn domain-invariant embeddings via a minimax game between a feature extractor and a domain classifier.

The vast majority of UDA research, including its theoretical foundations, focuses on classification tasks (Redko et al., 2022; Ben-David et al., 2010; Liu et al., 2022), with some works on regression tasks (Cortes & Mohri, 2014; Mansour et al., 2009) and only a few on generative tasks (Uppaal et al., 2024). More

Table 1: List of central acronyms, ordered by first appearance in text.

| Acronym | Definition |
| --- | --- |
| ABI | Amortized Bayesian Inference |
| NPE | Neural Posterior Estimation |
| UDA | Unsupervised Domain Adaptation |
| MMD | Maximum Mean Discrepancy |
| DANN | Domain-Adversarial Neural Networks |
| SBI | Simulation-Based Inference |
| MCMC | Markov Chain Monte Carlo |
| NLL | Negative Log Likelihood |
| NRMSE | Normalized Root Mean Squared Error |
| ECE | Expected Calibration Error |
| PC | Posterior Contraction |
| PPD | Posterior Predictive Distance |
| SSDD | Summary Space Domain Distance |
| C2ST | Classifier Two-Sample Test |
| INLD | Inference Network Latent Distance |
| IAT | Implicit Association Test |

recently, UDA methods have been successfully applied to address simulation-to-reality (sim2real) problems (Ćiprijanović et al., 2020; Swierc et al., 2023; Kong et al., 2023) which seek to generalize patterns learned in a simulated source domain to a real-world target domain. These problems seem pertinent to any simulation-based method relying on data generation from imperfect models.

**From Simulated to Real Domains** The preceding discussion makes the connection between UDA and ABI immediately apparent: When the distance between the data distribution $p(\boldsymbol{x}_{\mathrm{obs}})$ and the model-implied distribution $p(\boldsymbol{x}) = \mathbb{E}_{p(\boldsymbol{\theta})}\left[p(\boldsymbol{x} \mid \boldsymbol{\theta})\right]$ is non-zero, the risk of extrapolation error for atypical data $\boldsymbol{x}_{\mathrm{obs}}$ may increase. Indeed, this behavior has been observed repeatedly in the context of SBI (Ward et al., 2022; Schmitt et al., 2024; Huang et al., 2023; Frazier et al., 2024; Kelly et al., 2025). In particular, Frazier et al. (2024) notes that ABI is especially prone to "extrapolation bias" for observed summary statistics $\phi(\boldsymbol{x})$ that are far in the tails of the model-implied (i.e., prior predictive) density $p(\boldsymbol{x})$.

The scenario can be equivalently stated by invoking the notion of a *typical set* (Cover & Thomas, 2012), which denotes a subset of the support of $p(\boldsymbol{x})$ where most of the probability mass concentrates around the entropy $H(p)$:

$$A_\epsilon = \left\{\boldsymbol{x} \in \mathcal{X} : |-\log p(\boldsymbol{x}) - H(p)| \leq \epsilon\right\}. \tag{2}$$

Accordingly, for any problem-specific $\epsilon$, observed data $\boldsymbol{x}_{\mathrm{obs}} \notin A_\epsilon$ may result in a biased posterior approximation $q(\boldsymbol{\theta} \mid \boldsymbol{x}_{\mathrm{obs}})$. As further noted in the comprehensive theoretical exposition by Frazier et al. (2024), matching summary statistics $\phi(\boldsymbol{x}_{\mathrm{obs}})$ to the model-implied distribution of $\phi(\boldsymbol{x})$ can be a useful heuristic for reducing extrapolation bias. This observation harmonizes with the UDA literature as well (Ben-David et al., 2010). Pre-asymptotically, the success of such matching depends on multiple factors, including (i) the type and hyperparameters of the matching method (see Figure 2); (ii) the degree and nature of domain mismatch; (iii) the complexity of the learning problem; and (iv) even the choice of success metric. Thus, a primary goal of this work is to systematically examine the effects of these factors on a variety of metrics that can index potential robustness gains.

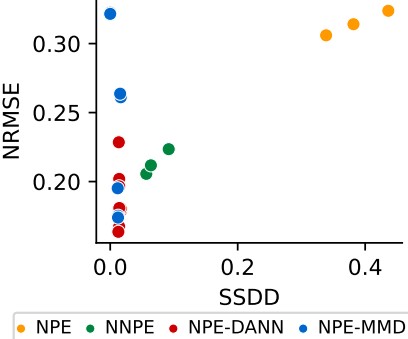

Figure 2: **Experiment 3**. Summary space domain distance (SSDD; MMD) vs. normalized root mean squared error (NRMSE) for row deletions. We observe a sweet spot of domain alignment without losing important information.

# 3 Methods

## 3.1 Unsupervised Domain Adaptation for Amortized Bayesian Inference

We start with the observation that model misspecification in ABI (Schmitt et al., 2024), and also more generally in neural SBI, can naturally be framed as an UDA problem: Ground-truth parameter values are only available for the simulated source domain $\mathcal{D} = \{(\boldsymbol{x}^i, \boldsymbol{\theta}^i)\}_{i=1}^N$ but not the observed target domain $\mathcal{D}_{\text{obs}} = \{\boldsymbol{x}_{\text{obs}}^i\}_{i=1}^{N_{\text{obs}}}$. In most machine learning applications, the collection of reliable ground-truth values is costly but feasible, whereas in SBI, collecting ground-truth parameter values $\boldsymbol{\theta}_{\text{obs}}$ of observed data is typically impossible. A general optimization objective for NPE-UDA methods can be formulated by extending the standard negative log-posterior NPE objective:

$$\mathcal{L}_{\text{NPE-UDA}}(q, \phi) := \mathcal{L}_{\text{NPE}} + \lambda \cdot \mathcal{L}_{\text{UDA}} \tag{3}$$

$$= \mathbb{E}_{p(\boldsymbol{\theta}, \boldsymbol{x})p(\boldsymbol{x}_{\text{obs}})}\big[-\log q(\boldsymbol{\theta} \,|\, \phi(\boldsymbol{x})) + \lambda \cdot d(\phi(\boldsymbol{x}), \phi(\boldsymbol{x}_{\text{obs}}))\big], \tag{4}$$

where $\lambda$ controls the regularization weight of the UDA loss and $d(\cdot, \cdot)$ is a divergence measure that attains its global minimum if and only if $\phi(\boldsymbol{x}) = \phi(\boldsymbol{x}_{\text{obs}})$.

$\mathcal{L}_{\text{NPE-UDA}}$ incurs a trade-off between approximation performance in the simulated domain and domain difference in the summary space, depending on the degree of domain mismatch. In the well-specified case, $p(\boldsymbol{x}) = p(\boldsymbol{x}_{\text{obs}})$, $\mathcal{L}_{\text{NPE-UDA}}$ reduces to the standard NPE loss. In the misspecified case, $p(\boldsymbol{x}) \neq p(\boldsymbol{x}_{\text{obs}})$, the summary network $\phi$ optimizes the summary statistics to both *maximize information extraction* in the simulated domain and *minimize domain shift* in summary space. Thereby, the inference network $q(\boldsymbol{\theta} \,|\, \phi(\boldsymbol{x}))$ needs to rely on domain-invariant information shared between the simulated and the observed domain.

The common UDA assumption that there exists a low-error hypothesis for both domains (Redko et al., 2022, cf. Eq.1) suggests a $\lambda$-dependent upper bound on the amount of domain shift that can be rectified by NPE-UDA methods. However, finding an optimal value for $\lambda$ despite missing target labels (i.e., ground-truth parameters in SBI) at test time is an open problem in UDA research (Zellinger et al., 2021; Musgrave et al., 2022). Thus, we expect this problem to carry over to NPE-UDA methods as well. Next, we formulate two NPE-UDA variants based on popular UDA methods with strong performance on established benchmarks (Musgrave et al., 2021).

## 3.2 NPE-MMD

The maximum mean discrepancy (MMD; Gretton et al., 2012) is a popular probability integral metric in SBI, since it can be efficiently estimated from a finite number of samples (Bischoff et al., 2024; Schmitt et al., 2024). For the same reason, it has been employed by various UDA works (Pan et al., 2010; Tzeng et al., 2014; Long et al., 2015) to measure the divergence between (transformed) samples from different domains. We categorize the combination of NPE and UDA based on MMD, such as the variants of Huang et al. (2023) and Swierc et al. (2024), as NPE-MMD. Choosing the MMD as $\mathcal{L}_{\text{UDA}}$, Eq. 3 becomes

$$\mathcal{L}_{\text{NPE-MMD}}(q, \phi) := \mathbb{E}_{p(\boldsymbol{\theta}, \boldsymbol{x})}\big[-\log q(\boldsymbol{\theta} \,|\, \phi(\boldsymbol{x}))\big] + \lambda \cdot \text{MMD}^2\big[\phi(\boldsymbol{x}) \,\|\, \phi(\boldsymbol{x}_{\text{obs}})\big]. \tag{5}$$

The most important hyperparameter of NPE-MMD is the choice of kernel in the sample-based MMD estimator. In our experiments, using a sum of inverse multiquadric kernels (Ardizzone et al., 2018) led to the most stable training dynamics, but other choices have been explored in the context of robust ABI as well, such as (sums of) Gaussian kernels (Schmitt et al., 2024; Huang et al., 2023).

## 3.3 NPE-DANN

Domain-adversarial neural networks (DANN; Ganin et al., 2016), which have not been considered for NPE to date, introduce a domain classifier $\psi(\cdot)$ to reduce domain distance. Unlike typical adversarial training, which alternates between objectives, DANN achieves minimax optimization in a single-step update via a gradient reversal layer (Ganin et al., 2016). This layer flips the gradient sign from the classifier to the feature extractor (e.g., summary network) $\phi$ during backpropagation, encouraging the feature extractor to

generate less domain-specific summary statistics. Similarly to NPE-MMD, DANN can be integrated into Eq. 3 to achieve NPE-DANN:

$$\mathcal{L}_{\text{NPE-DANN}}(q, \phi, \psi) := \mathbb{E}_{p(\boldsymbol{\theta}, \boldsymbol{x})}\big[ -\log q(\boldsymbol{\theta} \mid \phi(\boldsymbol{x}))\big] + \lambda \cdot \mathcal{L}_D(\psi, \phi). \tag{6}$$

The discriminator loss $\mathcal{L}_D$ is given by:

$$\mathcal{L}_D(\psi, \phi) := -\mathbb{E}_{p(\boldsymbol{x})}\big[\log(p(\psi(\phi(\boldsymbol{x}))))\big] - \mathbb{E}_{p(\boldsymbol{x}_{\text{obs}})}\big[\log(1 - p(\psi(\phi(\boldsymbol{x}_{\text{obs}}))))\big], \tag{7}$$

where $\psi$ is the domain classifier and the equation represents the binary cross-entropy loss on the domains, where a gradient reversal layer enables updating $\phi$ and $\psi$ in opposing directions.

While DANN is a powerful and popular UDA method (Zhou et al., 2022), it has two important drawbacks. First, the unstable training dynamics and convergence issues generally associated with adversarial learning can also occur with DANN (Sener et al., 2016; Sun et al., 2019). Second, adversarial training adds new hyperparameters, including the domain classifier architecture, an optional weight for gradient reversal balance (Ganin et al., 2016), and stabilization techniques like label smoothing (Zhang et al., 2023). Notably, although $\lambda$ is a shared hyperparameter in NPE-MMD and NPE-DANN, its effect on training dynamics will vary across applications due to differing $\mathcal{L}_{\text{UDA}}$ scales.

### 3.4 What Is the Target of Robustness?

To better understand the strengths and limitations of robust methods, including NPE-UDA, we suggest distinguishing between the following inference goals:

- **Target 1**: The analytic (true) posterior $p(\boldsymbol{\theta} \mid \boldsymbol{x}_{\text{obs}}) \propto p(\boldsymbol{x}_{\text{obs}} \mid \boldsymbol{\theta})\, p(\boldsymbol{\theta})$ of the assumed probabilistic model given the observed data $\boldsymbol{x}_{\text{obs}}$.

- **Target 2**: A posterior $p(\boldsymbol{\theta} \mid \widetilde{\boldsymbol{x}}_{\text{obs}}) \propto p(\widetilde{\boldsymbol{x}}_{\text{obs}} \mid \boldsymbol{\theta})\, p(\boldsymbol{\theta})$ of the assumed probabilistic model given *adjusted data* $\widetilde{\boldsymbol{x}}_{\text{obs}}$.

- **Target 3**: A posterior $\widetilde{p}(\boldsymbol{\theta} \mid \boldsymbol{x}_{\text{obs}}) \propto p(\boldsymbol{x}_{\text{obs}} \mid \boldsymbol{\theta})\, \widetilde{p}(\boldsymbol{\theta})$ from an *adjusted prior* $\widetilde{p}(\boldsymbol{\theta})$ given the observed data $\boldsymbol{x}_{\text{obs}}$.

**Target 1** is the most common target in Bayesian inference. Classical approximation methods such as MCMC almost always consider this target (Carpenter et al., 2017). **Target 2**, an explicit deviation from the true posterior, is often targeted by methods that seek to improve the robustness of Bayesian inference (see 4). Their goal is to reduce the influence of unmodeled phenomena in $\boldsymbol{x}_{\text{obs}}$, such as additional noise or external contamination, by approximating a target posterior $p(\boldsymbol{\theta} \mid \widetilde{\boldsymbol{x}}_{\text{obs}})$ based on denoised or uncontaminated data $\widetilde{\boldsymbol{x}}_{\text{obs}}$. This can be achieved either explicitly, by transforming $\boldsymbol{x}_{\text{obs}}$ into $\widetilde{\boldsymbol{x}}_{\text{obs}}$, or implicitly, by using an *adjusted (implicit) likelihood* $\widetilde{p}(\boldsymbol{x}_{\text{obs}} \mid \boldsymbol{\theta})$.

Since **Target 2** implies ignoring parts of the data that are in disagreement with the assumed probabilistic model, we expect corresponding methods to perform worse under prior misspecification: When a data-generating parameter $\boldsymbol{\theta}^*$ is impossible or highly unlikely under the assumed prior, *ignoring conflicting information effectively reduces the amount of information available to counteract a poorly chosen prior.* Generalized Bayes approaches (Bissiri et al., 2016) also aim to reduce the influence of undesired parts of the data. They move away from the classical Bayes rule by replacing the likelihood with a loss function, which can be interpreted as an adjusted likelihood $\widetilde{p}(\boldsymbol{x}_{\text{obs}} \mid \boldsymbol{\theta})$ according to **Target 2**. Lastly, **Target 3** can directly reduce the impact of prior misspecification by adjusting the prior based on $\boldsymbol{x}_{\text{obs}}$. However, compared to **Target 2**, it is more challenging to conceptualize the desired target priors $\widetilde{p}(\boldsymbol{\theta})$ and posteriors $\widetilde{p}(\boldsymbol{\theta} \mid \boldsymbol{x}_{\text{obs}})$ under model misspecification.

Given this categorization, what is the target of NPE-UDA? Unsurprisingly, the classic NPE loss $\mathcal{L}_{\text{NPE}}$ aims at **Target 1**. In contrast, the additional $\mathcal{L}_{\text{UDA}}$ loss governs the alignment of the summary space between simulated and observed data, effectively adjusting the observed data seen by the model. Thus, $\mathcal{L}_{\text{UDA}}$ introduces a shift towards **Target 2**, with $\lambda$ governing its relative importance compared to **Target 1**. As

hypothesized above, methods aiming at **Target 2** may not perform well under prior misspecification, which is confirmed for the NPE-UDA methods throughout our experiments. While Huang et al. (2023) suggested that their NPE-MMD variant is robust to prior mean shift, this conclusion was based on a single tested $\boldsymbol{x}_{\text{obs}}$ and our comprehensive evaluation could not replicate the result.

In line with our hypothesis and empirical results, Huang et al. (2023) observed that increasing values of $\lambda$ encourage trading off the information content of $\boldsymbol{x}$ to minimize the domain distance in summary space, leading the posterior to converge to the assumed prior $p(\boldsymbol{\theta})$. Thus, the critical importance of the tunable hyperparameter $\lambda$ in UDA contexts (Zellinger et al., 2021) directly translates to ABI applications, where $\lambda$ controls a trade-off between *improving* approximation under likelihood misspecification and *degrading* approximation under prior misspecification.

## 4 Related Work

**Robust Neural SBI**   Robustness in neural SBI has become a rapidly growing area of research, with most approaches enhancing robustness for a single data set at the cost of amortization, e.g., due to additional MCMC runs or post-hoc corrections. The majority of these approaches focuses on **Target 2** by incorporating a misspecification model (Ward et al., 2022), shifting observed summary statistics with low support (Kelly et al., 2023), reducing the influence of unmodeled data shifts via generalized SBI (Gao et al., 2023), or using the single-data-set NPE-MMD variant previously discussed (Huang et al., 2023). Focusing on **Target 1**, Siahkoohi et al. (2023) highlighted the role of the inference network's latent space in domain shifts and proposed a latent space correction based on the observed data $\boldsymbol{x}_{\text{obs}}$. Differently, Wang et al. (2024) focus on **Target 3** by using an upfront ABC run to filter the part of the parameter space causing the highest discrepancy between $\boldsymbol{x}$ and $\boldsymbol{x}_{\text{obs}}$.

**Robust ABI**   In contrast, research on robustifying inference while retaining amortization has been sparse. Extending the scope of the training data via additive noise (Cranmer et al., 2020; Bernaerts et al., 2023), such as the spike-and-slab noise approach of noisy NPE (NNPE; Ward et al., 2022), can be seen as a light modification to the simulator-implied likelihood of **Target 2**, but requires strong assumptions about the corruption process. Wehenkel et al. (2024) also approach **Target 2** by framing domain shift as an optimal transport problem in summary space, but this requires *observed* "ground-truth" parameters $\boldsymbol{\theta}_{\text{obs}}^*$ that are difficult to obtain in most ABI settings. Swierc et al. (2024) provided evidence for the potential of NPE-MMD for robust ABI but focused their evaluation solely on a gravitational lensing application with synthetically added noise. Finally, Glöckler et al. (2023) proposed an efficient regularization technique that can increase robustness against adversarial attacks and attain more reliable performance under **Target 1**.

## 5 Experiments

In all experiments, we benchmark NPE-MMD and NPE-DANN against an NPE baseline as well as NNPE (Ward et al., 2022, see Appendix B.1 for further details). We acknowledge that NNPE was not the primary focus of Ward et al. (2022) and instead consider it an instantiation of a simple additive noise training modification (Cranmer et al., 2020; Bernaerts et al., 2023), the presumably most straightforward current approach for robustifying ABI in practice.

The two existing works on NPE-MMD approaches mainly evaluated performance against contamination (Huang et al., 2023), where a fraction of the sample is replaced with corrupted observations (Huber, 1981), or noise applied to all observations (Swierc et al., 2024). Both of these scenarios are cases of likelihood misspecification where ignoring noise is desirable (**Target 2**). That is, the inference target is the posterior $p(\boldsymbol{\theta} \mid \widetilde{\boldsymbol{x}})$ of the assumed probabilistic model given the uncontaminated or (implicitly) denoised data set $\widetilde{\boldsymbol{x}}$.

Experiment 1 introduces a canonical contamination setting, in which we explore the sensitivity of NPE-UDA methods to hyperparameters using Bayesian optimization. Afterwards, we expand the scope by comprehensively evaluating various likelihood/data and prior misspecification scenarios to obtain clearer insights into the strengths and limitations of the robust methods. Experiment 2 starts with a simple and controllable setting that allows for comparing the NPE-UDA methods not only against standard NPE and

NNPE (Ward et al., 2022), but also to the analytic posterior under **Target 1**. Experiment 3 explores whether the result patterns can be replicated in a challenging setting with a high-dimensional parameter space. Lastly, since we are ultimately interested in the robustness of NPE in genuine scientific applications, Experiment 4 tests the methods on a massive real-world data set of human decision-making (von Krause et al., 2022).

We evaluate a range of metrics to enable a holistic assessment of the compared methods:

- **Parameter space performance metrics**: (i) Negative log likelihood (NLL) as a standard measure of posterior density estimation (ii); normalized root mean squared error (NRMSE) to measure approximation error; (iii) expected calibration error (ECE) to measure the fidelity of credible intervalst; (iv) posterior contraction (PC) to measure information gain from prior to posterior.

- **Data space performance metrics**: (i) Posterior predictive distance (PPD) to the observed data, which is the standard approach but has the disadvantage that the observed data contains the noise that a robust method should ignore, and (ii) PPD to data *resimulated* from the ground-truth parameters, a modification that allows PPD to represent the distance to well-specified (e.g., denoised) data. In Experiment 4, we approximate the denoised reference via intensive data pre-processing.

- **Network space metrics**: (i) Summary space domain distance (SSDD) measuring domain alignment via MMD, (ii) SSDD via the classifier two-sample test (C2ST), and (iii) inference network latent distance (INLD) to the base distribution of the generative neural network (diagonal Gaussian in all our experiments), which has recently been highlighted as the central mediator of posterior errors (Siahkoohi et al., 2023).

Appendix B provides details on metrics, experimental setups, network architectures, training and evaluation procedures, as well as additional results. Code for reproducing all results from this paper is available at https://github.com/bayesflow-org/NPE-UDA.

### 5.1 Experiment 1 - Ricker: Hyperparameter Exploration

**Setup** Our first experiment explores the properties of amortized NPE-UDA methods in a classic contamination misspecification scenario. The inference task consists of inferring two parameters of the popular Ricker model of population dynamics (Wood, 2010; Ricker, 1954). Using the same setting, previous work by Huang et al. (2023) demonstrated the potential of non-amortized NPE-MMD, which specializes inference for a single seen test data set.

Here, we evaluate both NPE-MMD and NPE-DANN in an *amortized* setting on $N_{obs} = 1\,000$ *unseen* observed (contaminated) test data sets. All methods are trained on a low budget of $N = 5\,000$ uncontaminated training data sets. The NPE-UDA methods are trained with additional $N_{obs} = 1\,000$ unlabeled data sets from the observed domain, non-overlapping with the validation and test set.

To achieve a comprehensive evaluation of the trade-offs associated with the additional hyperparameters of the NPE-UDA methods, we optimize for the NLL on $N_{obs} = 1\,000$ contaminated validation data sets via Bayesian hyperparameter optimization (Akiba et al., 2019) using 50 separate training runs per method. The fixed training run budget per method automatically accounts for the complexity of the hyperparameter space, since fewer hyperparameters allow for a more thorough exploration of a method's hyperparameter space. We optimize the learning rate for all methods and the $\lambda$ hyperparameter controlling the alignment strength for NPE-MMD and NPE-DANN (see Table B.1 for the search ranges for all hyperparameters).

For NPE-DANN, we additionally optimize (i) the width and depth of the feedforward discriminator architecture; (ii) the weight $\lambda_{grl}$ balancing the strength of the adversarial summary network updates relative to the discriminator network (with $\lambda_{grl} < 1$ diminishing and $\lambda_{grl} > 1$ amplifying the reversed classification gradients passed to the summary network); and (iii) label smoothing, which has been found to be helpful for stabilizing the dynamics of domain-adversarial training (Zhang et al., 2023). [1]

---

[1] We also explored scaling the gradient reversal weight $\lambda_{grl}$ during training as suggested by Ganin et al. (2016), but found consistently worse results for the tested NPE setting. The same applies to the kernel choice in NPE-MMD, where the popular

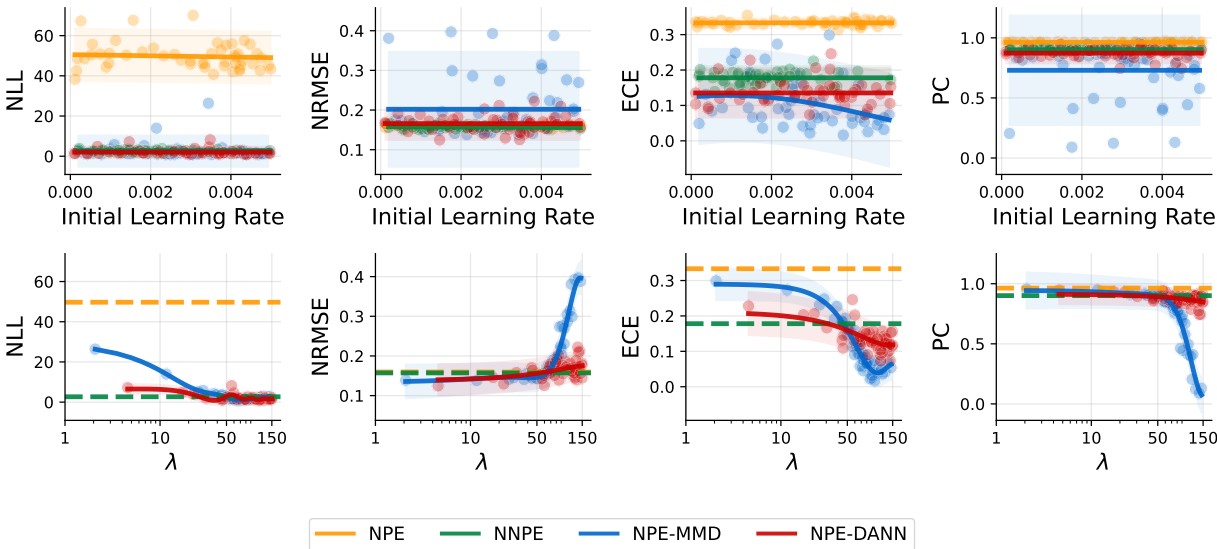

Figure 3: **Experiment 1**. Parameter space performance metrics resulting from 50 separate Bayesian hyper-parameter optimization runs per method. The solid trend lines represent the predictive mean of a Gaussian process regression fitted to the individual run results, with the shaded areas representing 95% confidence intervals of the predictive distribution. If a parameter was not optimized, the method's average performance is depicted by a dashed horizontal line. Lower values indicate better performance for all metrics but PC. NLL = Negative Log Likelihood. NRMSE = Normalized Root Mean Squared Error. ECE = Expected Calibration Error. PC = Posterior Contraction. Whereas learning rate optimization is mostly ineffective for improving performance under contamination misspecification, the domain alignment regularization parameter $\lambda$ controls a trade-off between error (NRMSE) vs. calibration (ECE) and contraction (PC) for NPE-UDA methods.

**Results** The parameter space performance metrics results are displayed in Figure 3, the data space performance metrics results in Figure 4a, and the network space metrics in Figure 4b. Overall, optimizing the learning rate does not improve approximation performance on contaminated data, with the only exception of NPE-MMD calibration improving at higher learning rates (see Figure 3). As expected, NNPE exhibits consistently robust performance as its data-generating process resembles the contamination misspecification scenario, whereas the success of the NPE-UDA methods depends on the domain alignment regularization parameter $\lambda$.

Concerning the parameter space performance metrics (Figure 3), we observe a lower NLL for all robust methods compared to NPE. However, taking into account the other metrics unveils that robust methods tend to improve calibration, but not estimation error. For the NPE-UDA methods, increasing the domain alignment regularization parameter $\lambda$ improves calibration at the cost of approximation error and posterior contraction. We also find NPE-MMD to be much more sensitive to $\lambda$, with inference breaking down at very large values.

With regard to data space performance metrics (Figure 4a), both approaches of obtaining the posterior predictive distance lead to similar results in this setting. While the PPD mostly mirrors the NRMSE in the parameter space, we find a surprising advantage of NNPE in the data space. Since we find a close correspondence between PPD and NRMSE in the other experiments, NNPE's unique advantage in the current setting could be caused by a peculiar sensitivity of the Ricker simulator to certain parameter constellations that NNPE is less likely to infer.

---

choice of using sums of Gaussian kernels with different bandwidths (Muandet et al., 2017; Schmitt et al., 2024) destabilized training compared to the sum of inverse multiquadric kernels. Thus, we did not include these hyperparameters in the systematic hyperparameter assessment.

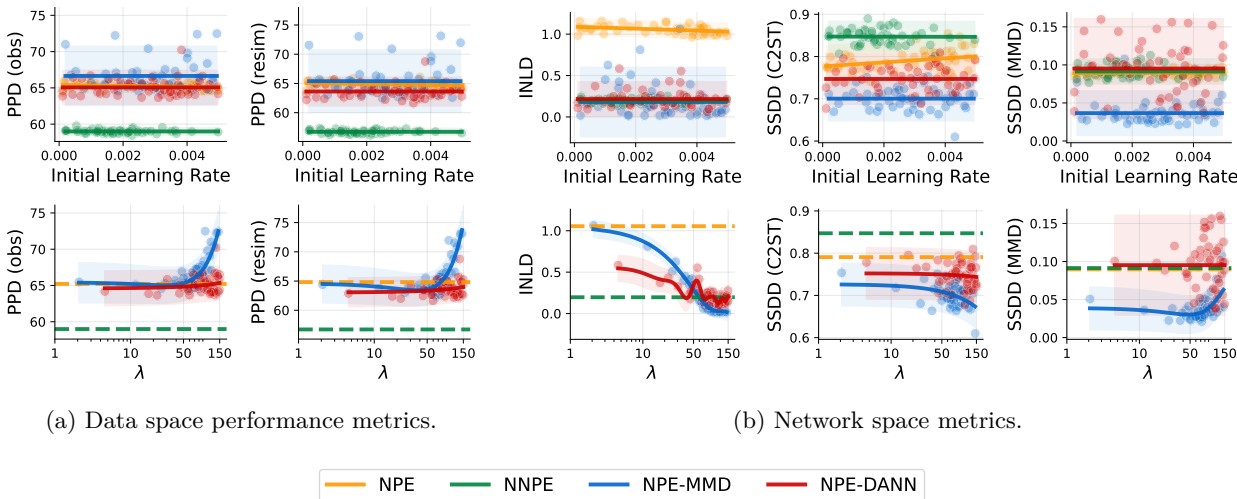

(a) Data space performance metrics.                    (b) Network space metrics.

Figure 4: **Experiment 1**. Further metrics resulting from 50 separate Bayesian hyperparameter optimization runs per method. The solid trend lines represent the predictive mean of a Gaussian process regression fitted to the individual run results, with the shaded areas representing 95% confidence intervals of the predictive distribution. If a parameter was not optimized, the method's average performance is depicted by a dashed horizontal line. Lower values indicate better performance for all metrics but SSDD. PPD = Posterior Predictive Distance (RMSE). INLD = Inference Network Latent Distance (MMD). SSDD = Summary Space Domain Distance.

Considering network space metrics (Figure 4b), we find that the deformation of the inference network's latent space, as measured by INLD, directly corresponds to density estimation quality as measured by NLL. This is unsurprising, given the change-of-variable mechanics of normalizing flows (Papamakarios et al., 2021). Further, we observe an overall lower summary space domain distance (SSDD) for both NPE-UDA methods in terms of C2ST but only for NPE-MMD in terms of MMD, and we do not observe the expected decrease in SSDD with increasing $\lambda$. We suspect these patterns to be caused by two factors limiting SSDD variability: First, the overall domain distance being relatively small and second, the hyperparameter optimization search concentrating on higher $\lambda$ values. Our next experiments inspect the lower range of $\lambda$ settings more closely and reveal it to be decisive for SSDD. Lastly, we find an overall little impact of the additional NPE-DANN hyperparameters (see Figure B1).

### 5.2 Experiment 2 - 2D Gaussian Means: Simple and Controllable Benchmark

**Setup** Inspired by Schmitt et al. (2024), our next experiment tests the performance of NPE-UDA methods against different *types* of misspecification. Here, the simple approximation task of inferring the means of a 2-dimensional Gaussian model enables a comparison to an analytic posterior, which represents the optimal solution under **Target 1**. The well-specified setting uses a multivariate standard normal prior and an identity likelihood covariance matrix. We evaluate performance under increasing misspecification in two prior misspecification scenarios—prior location $\boldsymbol{\mu_0}$ and prior scale $\boldsymbol{\Sigma_0} = \tau_0 \boldsymbol{I}_2$—and two likelihood misspecification scenarios—likelihood scale $\boldsymbol{\Sigma} = \tau \boldsymbol{I}_2$ and data contamination $\epsilon$ (see Table B.2). For the contamination misspecification, a fraction $\epsilon$ of the observations is replaced by negative and positive vectors of the constant $c = 1.5$ to obtain atypical observations without affecting overall location or scale. Each simulated data set contains $M = 100$ exchangeable observations. All methods train on $N = 48\,000$ well-specified data sets until convergence, with NPE-UDA methods additionally exposed to $N_{\text{obs}} = 48\,000$ unlabeled data sets. All methods are evaluated on $N_{\text{obs}} = 1\,000$ observed data sets (unseen by NPE-UDA methods).

**Results** Figure 5 displays the results for all misspecification scenarios. We invert the meaning of the posterior contraction (PC) metric, such that lower means better for all metrics. Thus, the performance of a

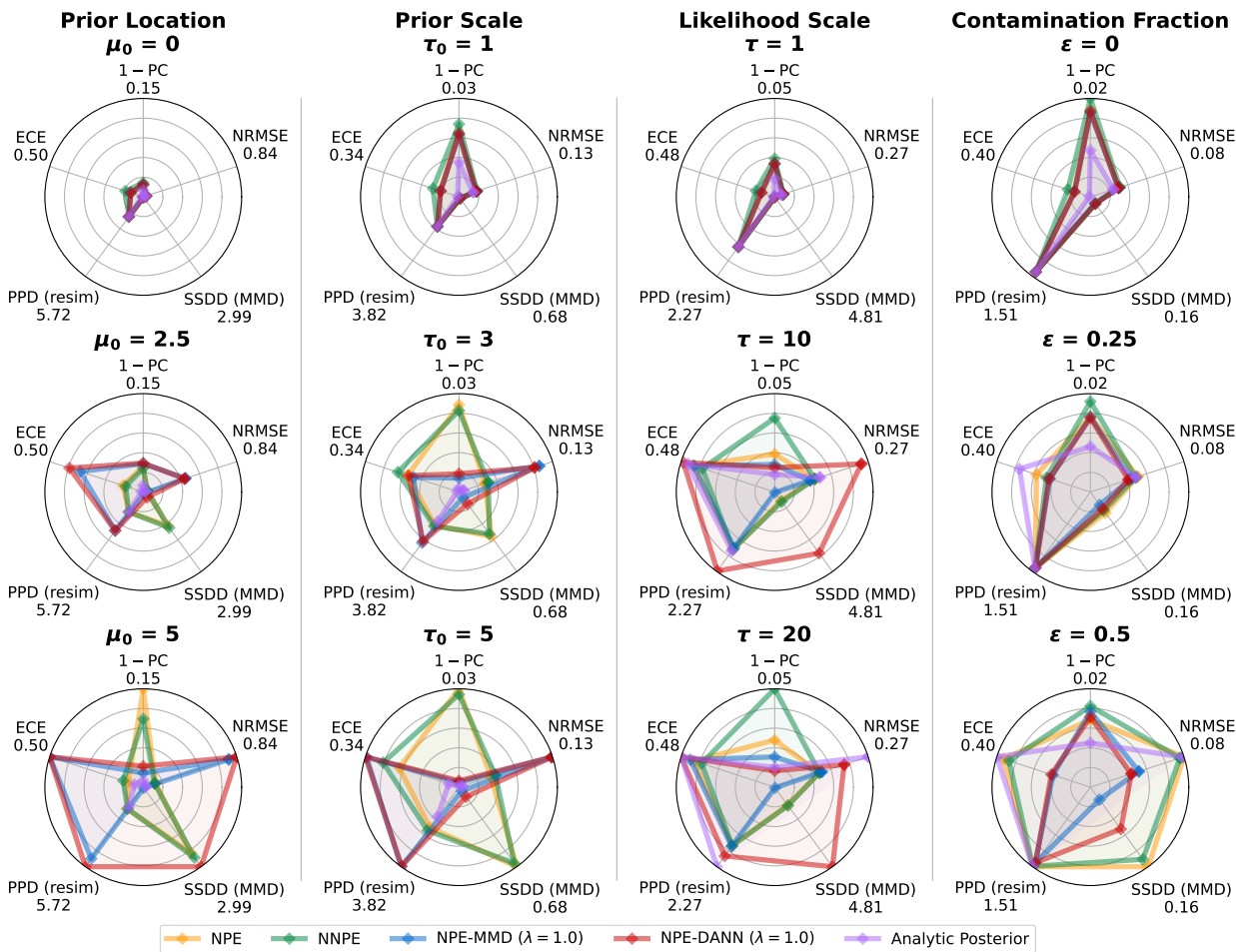

Figure 5: **Experiment 2**. Performance metrics and summary space domain distance (SSDD) of the methods in all misspecification scenarios (columns), aggregated via the median of 10 runs. The first row shows the well-specified setting, with misspecification increasing from top to bottom within each column. Metric values are centered at 0 and normalized by each column's/scenario's maximum value, which is displayed below the metric name at the border of each radar plot. Lower values indicate better performance for all metrics but SSDD. $1-$ PC $= 1-$ Posterior Contraction. NRMSE = Normalized Root Mean Squared Error. SSDD (MMD) = Summary Space Domain Distance measured via MMD (not applicable for Analytic Posterior). PPD (resim) = Posterior Predictive Distance measured via the RMSE to resimulated data. ECE = Expected Calibration Error. NPE-UDA methods fail under prior misspecification but can be advantageous under contamination.

method can mostly be inferred from its area. All methods perform well in the well-specified case (first row), whereas we observe distinct but consistent patterns for the different methods under increasing mismatch.

In the prior misspecification scenarios, NPE and NNPE perform well compared to the analytic posterior under a location shift, but fall off under a scale shift, with increasing error (NRMSE) and drastically increased miscalibration (ECE). The two NPE-UDA methods, on the other hand, perform poorly for both prior location and scale shift (see Section B.4 for the complementary prior scale shift scenario of too broadly chosen priors).

In the likelihood scale misspecification scenario, the NPE methods are generally less sensitive to the misspecification than the analytic posterior. NPE-MMD successfully aligns the summary space between domains, but the practical effects of this alignment are limited to slightly improved PC. NPE-DANN, on the other hand, shows the unstable training dynamics described in Section 3.3: It fails to align the summary space

for both likelihood scale misspecification levels (as well as the $\mu_0 = 5$ prior location shift scenario), which translates to poor performance. Crucially, *this drastic failure in the observed domain is not detectable in the simulated domain*, where all methods, even NPE-DANN in the $\tau = 20$ scenario, perform well according to all metrics and only SSDD signals irregularities (see Figure B3).

In the data contamination scenario, deviating from the true posterior via **Target 2** enables NPE-MMD and NPE-DANN to excel, achieving much lower NRMSE and ECE than NPE, NNPE, *and even the analytic posterior*. Across all misspecification scenarios, NNPE performs similarly to NPE, with NNPE's noisier training process typically resulting in slightly lower contraction and calibration. This applies even to the contamination scenario, where we expected an advantage for NNPE due to its similarity to the corruption process.

With regard to differences between metrics, PPD reliably detects NPE-UDA failures under prior misspecification, but is less sensitive under likelihood scale shifts and insensitive under contamination. We suspect that this lessened informativeness compared to Experiment 1 is caused by the simple data structure of the Gaussian mean model. Further, the deformation of the inference network's latent space, measured via INLD, is again strongly associated with approximation quality, with rank correlations of $r = .79$ with NRMSE, $r = .95$ with ECE, and $r = .74$ with PPD (re-simulated).

Furthermore, in this low-dimensional example, we observed two sources of instability for NPE-UDA methods. First, despite the good performance across a wide range of $\lambda$ settings in Experiment 1, we observed a high sensitivity to the $\lambda$ regularization hyperparameter. For $\lambda = 0.1$, the domain alignment in the contamination scenario is too weak, eliminating the advantage of NPE-UDA methods (see Figure B5). Differently, setting $\lambda = 10$ leads to drastic failures due to overly aggressive domain alignment of NPE-MMD and increases the training instabilities of NPE-DANN (see Figure B6). Second, these extreme failures necessitated aggregation of the training run results via the median instead of the mean, since single extreme results (not for $\lambda = 0.1$, occasionally for $\lambda = 1$, frequently for $\lambda = 10$) rendered visualization of the results impossible.

### 5.3 Experiment 3 - Bayesian Denoising: High-Dimensional Benchmark

**Setup** This experiment tests the generalization of our results on a high-dimensional (in the context of SBI) benchmark simulating a noisy camera model (Ramesh et al., 2022). The parameter vector $\boldsymbol{\theta} \in \mathbb{R}^{256}$ represents a crisp image, whereas the observation $\boldsymbol{x} \in \mathbb{R}^{256}$ is a blurred version of the original image generated by the noisy camera. The training data set consists of $N = 50\,000$ images from the MNIST data set (Lecun et al., 1998), downscaled to $16 \times 16$ pixels for compatibility with the USPS data set (Hull, 1994).

We test four different misspecification scenarios (see Table 3 for examples). In the prior misspecification scenario, we keep the settings of the noisy camera model constant but use images from the USPS data set (Hull, 1994). While both data sets contain digits, the USPS data set features smaller margins, giving the priors different support. In the likelihood scale scenario, we increase the amount of blur. In the noise contamination scenario, we replace 10% of the pixels with salt-and-pepper noise (i.e., set them to black or white). In the row contamination scenario, we randomly set two rows (12.5% of the pixels) of each observation to black. The NPE-UDA methods are trained with $N_{\text{obs}} = 1\,000$ observed training data sets. We evaluate the performance on further $N_{\text{obs}} = 1\,000$ observed test data sets (unseen by the NPE-UDA methods during training).

**Results** Table 2 displays an overview of the metrics in all scenarios. Table 3 shows samples from the best run (lowest NRMSE) for each scenario and method. We observe worse approximations for all robust methods compared to NPE in the prior misspecification scenario, even though the summary space domain distance (SSDD) is strongly diminished for NPE-DANN and NPE-MMD. This is somewhat expected, as performance improvements would also require an adaptation of the inference network, which cannot be induced by the methods tested here. NNPE is beneficial in the two contamination scenarios, whereas NPE-DANN and NPE-MMD improve performance in all three likelihood misspecification scenarios. The results highlight the differences between the robust methods: While NNPE mainly excels in the noise contamination scenario, where its misspecification model matches the domain shift, NPE-UDA methods effectively adapt to different likelihood shifts. Figure 6 shows a two-dimensional t-SNE (Van der Maaten & Hinton, 2008) representation

| Method | $\lambda$ | Prior (MNIST → USPS) | | | Likelihood Scale | | | Contamination (Noise) | | | Contamination (Rows) | | |
|---|---|---|---|---|---|---|---|---|---|---|---|---|---|
| | | NRMSE ↓ | PPD ↓ | SSDD | NRMSE ↓ | PPD ↓ | SSDD | NRMSE ↓ | PPD ↓ | SSDD | NRMSE ↓ | PPD ↓ | SSDD |
| NPE | - | **0.259** | **0.131** | 0.256 | 0.165 | 0.036 | 0.101 | 0.312 | 0.117 | 0.463 | 0.315 | 0.112 | 0.386 |
| NNPE | - | 0.274 | 0.137 | 0.206 | 0.173 | 0.041 | 0.088 | **0.154** | **0.035** | 0.019 | 0.214 | 0.064 | 0.071 |
| NPE-DANN | 0.01 | 0.329 | 0.193 | 0.027 | **0.130** | 0.031 | 0.027 | 0.217 | 0.065 | 0.017 | 0.180 | 0.056 | 0.016 |
| NPE-DANN | 0.10 | 0.326 | 0.193 | 0.029 | 0.134 | 0.031 | 0.016 | 0.211 | 0.057 | 0.015 | **0.178** | **0.045** | 0.014 |
| NPE-DANN | 1.00 | 0.352 | 0.205 | 0.038 | 0.147 | 0.032 | 0.014 | 0.240 | 0.068 | 0.015 | 0.201 | 0.051 | 0.014 |
| NPE-MMD | 0.01 | 0.303 | 0.184 | 0.029 | 0.145 | **0.027** | 0.022 | 0.268 | 0.085 | 0.016 | 0.262 | 0.085 | 0.016 |
| NPE-MMD | 0.10 | 0.312 | 0.189 | 0.018 | 0.189 | 0.059 | 0.009 | 0.245 | 0.071 | 0.013 | 0.181 | 0.047 | 0.012 |
| NPE-MMD | 1.00 | 0.374 | 0.225 | 0.004 | 0.322 | 0.129 | 0.000 | 0.321 | 0.129 | 0.000 | 0.322 | 0.129 | 0.000 |

Table 2: **Experiment 3**. Metrics of the methods in all misspecification scenarios, averaged across 3 runs. NRMSE: Normalized Root Mean Squared Error (lower is better). PPD = Posterior Predictive Distance (RMSE) to resimulated data (lower is better). SSDD = Summary Space Domain Distance (MMD). Lower values indicate better summary space alignment, but too much alignment (i.e., vanishing SSDD) can lead to an uninformative summary space (e.g., NPE-MMD with $\lambda = 1.00$).

Table 3: **Experiment 3**. Parameters, observations, and samples from the run with the lowest NRMSE for each scenario and method. *Train* shows a sample from the parameters $\boldsymbol{\theta}$ of the training distribution and the corresponding observations $\boldsymbol{x}_{\text{obs}}$. The observations are identical for NPE, NPE-DANN, and NPE-MMD, whereas spike-and-slab noise is added for NNPE. The similarity to the observations in the *Contamination (Noise)* scenario explains the good performance of NNPE in that scenario.

of the summary spaces produced by the different methods. The observed overlap corresponds well to the SSDD values reported in Table 2.

Overall, NPE-DANN achieves good performance over a wide range of $\lambda$ values. In contrast, NPE-MMD is prone to overregularizing the summary space, leading to a complete loss of information in the summary space (see also Figure 2). This is indicated by a huge drop in performance and vanishing SSDD. We found NPE-MMD highly sensitive to the chosen batch size, which we had to increase from 32 to 128 to achieve acceptable results. Thus, increasing the batch size and reducing $\lambda$ can counteract excessive regularization in higher-dimensional problems. Finally, the close correspondence between the NRMSE and PPD metrics confirms our hypothesis that the limited diagnostic power of PPD in the likelihood misspecification scenarios of Experiment 2 was caused by the limited informativeness of data simulated from a simple Gaussian model.

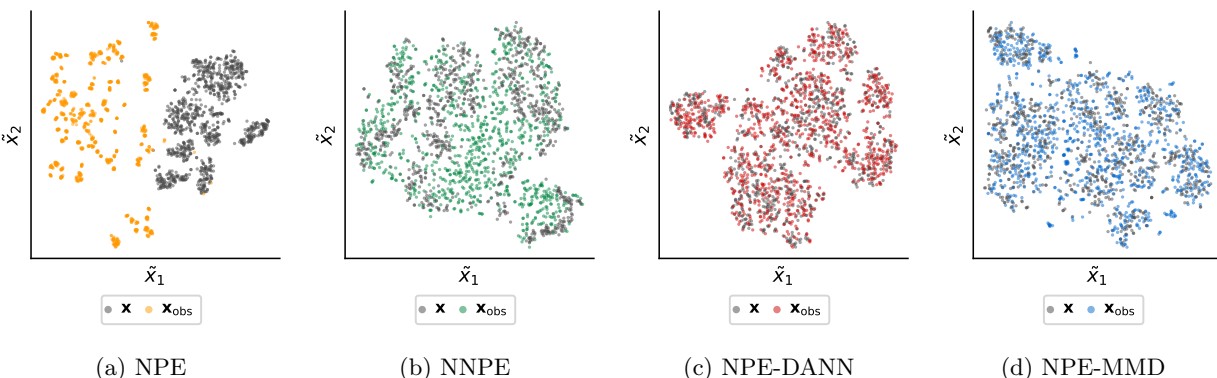

Figure 6: **Experiment 3**. Contamination (Rows): t-SNE representation of the summary spaces from the best run (lowest NRMSE) of each method. The t-SNE map is calculated jointly using summary statistics of data from both the simulated and the observed domain. For NPE, the two domains are clearly separated. With increasing alignment in the summary space, the overlap between the domains increases: The domain embeddings overlap partially for NNPE (medium SSDD in Table 2) and fully for NPE-DANN and NPE-MMD (low SSDD in Table 2). Since t-SNE can introduce artificial clustering, the overlap and not the specific shape is relevant.

## 5.4 Experiment 4 - Decision Making: Large Human Behavior Data Set

**Setup** Finally, we evaluate the effectiveness of NPE-UDA methods using a real-world example from cognitive science, based on a large-scale empirical data set of a binary decision-making task. The data set comprises response time data from millions of participants who completed the Implicit Association Test (IAT), a widely used test in social psychology (Greenwald et al., 1998; 2003). The data are publicly available through Project Implicit (Xu et al., 2014).

To gain insights into the underlying cognitive processes, response time data are often analyzed using evidence accumulation models (Evans & Wagenmakers, 2020; Ratcliff et al., 2016). However, fitting these models within a Bayesian framework is computationally intensive, making them a prime use case for ABI. A further challenge with online response time data is the presence of substantial noise, stemming from limited experimental control (Gong & Huskey, 2023). For example, participants may guess, experience attentional lapses, or behave inconsistently. This issue can be framed as a form of likelihood misspecification, where the observed data are partially generated by processes not captured by the model.

Traditional approaches to address this include data cleaning procedures or extending the model with additional parameters (e.g., to capture guessing behavior, see Ratcliff & Tuerlinckx, 2002). Here, we explore whether NPE-UDA methods can offer an alternative by aligning the summary statistics of simulated (clean) model data with those of the noisy empirical data. This alignment should allow for improved parameter estimation despite the noise. Thus, we aim for **Target 2**: approximating the posterior distributions of cognitive model parameters while implicitly filtering out the noise components in the empirical data. All methods are trained on $N = 32\,000$ simulated response time data sets to approximate the 6 parameters of an evidence accumulation model adapted to IAT task data. The NPE-UDA approaches additionally train on $N_{\text{obs}} = 32\,000$ unprocessed empirical data sets.

For evaluation, we construct several empirical test data sets using held-out observed data sets along two dimensions (see Appendix B.6 for details): (i) data are either unprocessed or cleaned using gold standard pre-processing procedures from the cognitive modeling literature, and (ii) data are categorized as either well-specified — where the cognitive model is expected to be valid — or misspecified, representing cases likely affected by processes not captured by the model (i.e., likelihood misspecification) based on their atypicality in NPE summary space. The misspecified test set consists of data from $N_{\text{obs}} = 730$ participants, while the well-specified test set consists of $N_{\text{obs}} = 10\,000$ randomly selected participants for whose data the probabilistic model is classified as well-specified.

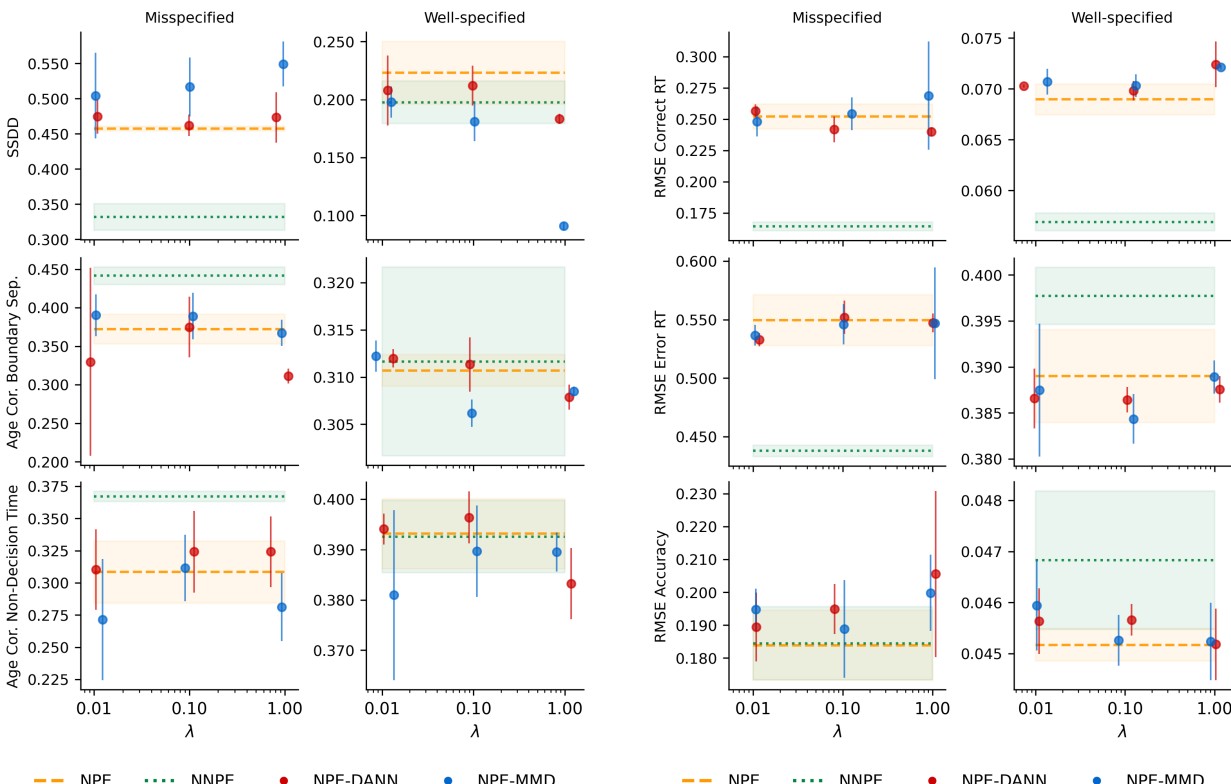

(a) Summary space domain distances (SSDDs; MMD) and external validity metrics for empirical data. The first row shows the SSDDs of simulated vs. empirical data. Rows two and three show (across-person) age correlations for posterior medians in two cognitive model parameters: boundary separation (in the congruent experimental condition) and non-decision times (for correct trials).

(b) Posterior predictive distances (PPDs; RMSE) to cleaned empirical response time (RT) data. The first row shows the PPD for response times (in seconds) on correct trials, averaged across posterior samples, participants, response time quantiles, and experimental conditions. The second row shows the PPDs for error response times, while the third row shows the PPDs for accuracy rates (in %).

Figure 7: **Experiment 4**. Metric results for all methods and different $\lambda$ weights for the NPE-UDA methods. All runs display the averaged results across three runs per method, with across-run standard deviations shown as shaded areas (for NPE and NNPE) or error bars (for NPE-DANN and NPE-MMD). Please note that y-axis scales differ between the misspecified and the well-specified settings. $\lambda$ = UDA regularization weight. The left column shows results for misspecified data sets, while the right column shows results for the (much larger) well-specified data set. While NNPE improves performance on misspecified data sets, the adaptation of NPE-UDA methods to the general observed domain does not cover the minority of misspecified data sets.

To evaluate the networks' performance, we first compare the methods by assessing the summary space domain distances between simulated and empirical data sets. Next, because certain cognitive parameters are known to covary with participant age (von Krause et al., 2020; Ratcliff et al., 2010; Theisen et al., 2021), we then examine how well each method captures this relationship by comparing age correlations for two key parameters across estimation approaches. Finally, we use the PPD to assess the methods based on their ability to predict unseen empirical data. Similar to before, we use a denoised version of the data as PPD reference, which we approximate for this real-world application via the intensive pre-processing procedures.

We train the NPE-UDA methods with $\lambda$ weights of 0.01, 0.1, 1, and 10. As before, $\lambda = 10$ frequently caused training instabilities (particularly for the MMD approach) leading to extreme results. To retain readable visualizations, we leave out the $\lambda = 10$ failure setting in the following.

**Results** Figure 7 presents the main results from our decision modeling experiment. Concerning Figure 7a, the NPE-UDA methods show the expected reduction in SSDD in the well-specified data sets, with domain alignment becoming more pronounced as the UDA weight parameter $\lambda$ increases. However, this domain adaptation effect does not extend to the minority of misspecified test sets, where NNPE shows the smallest distance between domains in summary space. With respect to age correlations for the two cognitive parameters, NNPE leads to on average slightly higher correlations on misspecified data compared to the other approaches.

Figure 7b presents posterior retrodictive RMSE values between posterior resimulations and the cleaned test data; corresponding results for the uncleaned data are provided in the appendix (Figure B14). Consistent with the SSDD and correlation results, we do not observe clear advantages for any method on the well-specified data sets but an advantage of NNPE on the misspecified data sets for response time fits. All methods perform comparably concerning the accuracy fits. The NPE-UDA approaches perform similarly to NPE (cf. Figure 7b), with higher $\lambda$ values leading to increased instability as indicated by RMSE variability across runs for misspecified data. Overall, while additive training noise via the NNPE method proved beneficial for misspecified data sets, the general domain adaptation induced by the NPE-UDA methods did not carry over from the majority of well-specified data sets to the minority of misspecified data sets.

## 6    Discussion

**NPE-UDA Methods** In this paper, we argued that introducing UDA to NPE methods shifts the inference target from the standard analytic posterior $p(\boldsymbol{\theta} \mid \boldsymbol{x}_{\mathrm{obs}})$ to a "denoised posterior" $p(\boldsymbol{\theta} \mid \widetilde{\boldsymbol{x}}_{\mathrm{obs}})$ based on adjusted data $\widetilde{\boldsymbol{x}}_{\mathrm{obs}}$. This shift implies potential robustness gains under likelihood misspecification, where implicitly ignoring unmodeled phenomena in the observed data can be desirable. However, it also reduces the amount of information available to counteract prior misspecification. We consistently found these patterns throughout our systematic evaluations for both the existing NPE-MMD and our new NPE-DANN method. Consequently, prior predictive or pushforward checks (Bürkner et al., 2025; Box, 1980) become an even more essential part of the Bayesian workflow (Gelman et al., 2020) for NPE-UDA methods.

Our results suggest that while posterior accuracy is related to domain distance in the summary space, the relationship is not straightforward, pointing to more subtle effects of domain alignment. Compared to simpler methods, such as NNPE (Ward et al., 2022), the flexibility of NPE-UDA methods to automatically adapt to various types of likelihood misspecification comes at the cost of reduced transparency during inference. Thus, a closer examination of UDA mechanisms is necessary to determine the nature of domain adaptation and how exactly these adaptations affect the interpretation of the resulting posterior $p(\boldsymbol{\theta} \mid \widetilde{\boldsymbol{x}}_{\mathrm{obs}})$. Employing interpretability methods or decoders to track summary space adaptations back to the data space seems a particularly promising avenue for future research.

**Role of the Regularization Weight** NPE-UDA methods, and especially NPE-DANN, introduce a number of additional hyperparameters that could potentially influence inferential results. While we found limited effects of most hyperparameters in our hyperparameter exploration experiment, we confirmed the existence of an application-specific optimal amount of UDA regularization (Zellinger et al., 2021) in the NPE context across all experiments. Both NPE-UDA methods exhibited substantial instabilities for higher $\lambda$ values: We observed the typical unstable training dynamics associated with adversarial training for NPE-DANN, whereas NPE-MMD exhibited overly aggressive domain alignment, suppressing all information contained in the data. Notably, which $\lambda$ values qualify as high varied substantially between the experiments: while a broad range of $\lambda$ values was tolerable in Experiment 1, we found severe inference breakdowns already for $\lambda = 10$ in all other experiments.

**Measuring Robustness in the Real World** The critical influence of the $\lambda$ hyperparameter directly leads to the next question: How can we measure robustness gains in the real world, where ground-truth parameter values are unavailable, and find an optimal value for $\lambda$? Posterior predictive measures that compare posterior resimulations to the observed data are usually the tool of choice. For measuring the success of robust methods, however, their standard application is flawed, since using the observed data directly as a reference implies that successfully ignoring noise is *punished* by increased posterior predictive distance to noisy outliers. Our

evaluation metrics sought to account for this by constructing a "denoised" reference data set. Although this approach is useful for benchmarking robust methods on real-world data, creating application-specific reference data solely for hyperparameter tuning would negate the benefit of automatic domain adaptation. Alternative approaches to measuring the robustness of NPE in the real world are also challenging to realize for NPE-UDA methods: The summary space is actively reshaped by NPE-UDA methods and therefore loses its validity as a proxy for detecting model misspecification (Schmitt et al., 2024). In contrast, the predictive variability of a deep ensemble (Lakshminarayanan et al., 2017) on the observed data, which directly measures the predictive fragility resulting from model misspecification in NPE (Elsemüller et al., 2024), remains a meaningful diagnostic for NPE-UDA. However, using deep ensemble variability to guide hyperparameter search is problematic: Ensemble variability can straightforwardly be minimized by collapsing the posterior to the prior, thus rewarding excessive regularization during hyperparameter search. While such collapses could be partially addressed by jointly optimizing hyperparameter search for posterior contraction, doing so introduces the additional challenge of balancing these two search objectives in label-free real-world settings. Furthermore, embedding deep ensembles into hyperparameter searches multiplies the already large computational cost of the latter and consequently remains impractical. Thus, finding generally reliable robustness measures for real-world settings remains an unsolved issue, embedded into the overarching open UDA problem of guiding hyperparameter optimization despite missing target labels (Zellinger et al., 2021; Musgrave et al., 2022).

**Future Avenues**   Based on the results of our experiments, we believe two further pathways to be especially interesting for future research. First, we focused on UDA approaches that target the summary space to retain the modularity of summary/inference network NPE architectures. This directly enables extensions to different downstream tasks, such as amortized Bayesian model comparison (Radev et al., 2021; Elsemüller et al., 2023; Jeffrey & Wandelt, 2024). The latter reframes model comparison as a classification task and is thus situated in the most extensively studied UDA task setting. Second, in contrast to the non-amortized NPE-MMD approach by Huang et al. (2023), which specializes inference for a single observed data set, we evaluated amortized approximation performance on data sets *unseen* by the NPE-UDA methods. While we observed good performance with low amounts of observed training data, it would be valuable to systematically assess the minimum amount of data necessary for effective adaptation to the observed domain. Here, generalization gaps of NPE-UDA methods could be measured via the difference in performance on data sets seen and unseen during training.

**Conclusion**   Our results reveal a rather complex story of NPE-UDA shaped by the interplay of factors such as problem dimensionality, misspecification type, and UDA model choice. They also emphasize the need for pairing stylized theoretical investigations with thorough empirical evaluation. In sum, UDA offers a straightforward way to incorporate real data into simulation-based training and shows promise in handling various types of likelihood misspecification. However, our systematic evaluation uncovered major obstacles and unanswered questions that hinder the direct application of NPE-UDA methods in critical settings.

**Acknowledgments**

The authors are grateful for the constructive feedback received during reviews and live discussions of an early version of this work at the "Frontiers in Probabilistic Inference: Learning meets Sampling" workshop at ICLR 2025. L.E. and V.P. are supported by the state of Baden-Württemberg through bwHPC. Additionally, L.E. is supported by a grant from the Deutsche Forschungsgemeinschaft (DFG, German Research Foundation; GRK 2277; project number 310365261) to the research training group Statistical Modeling in Psychology (SMiP). V.P. is supported by the Konrad Zuse School of Excellence in Learning and Intelligent Systems (ELIZA) through the DAAD programme Konrad Zuse Schools of Excellence in Artificial Intelligence, sponsored by the Federal Ministry of Education and Research, and the German Research Foundation (DFG) through grant INST 35/1597-1 FUGG. M.V.K. acknowledges funding by the German Research Foundation (DFG) through grant KR 6065/1-1. P.B. acknowledges support of DFG Project 528702768 and DFG Collaborative Research Center 391 (Spatio-Temporal Statistics for the Transition of Energy and Transport) – 520388526. S.T.R is supported by NSF under Grant No. 2448380.

**Author Contributions**

L.E. and S.T.R. conceived the initial idea, L.E., P.C.B., and S.T.R. developed the methodology, and A.V. and S.T.R. supervised the project. L.E., V.P., and M.V.K. implemented the experiments and wrote the experiment sections. L.E., P.C.B., and S.T.R. wrote the rest of the initial manuscript draft, which all authors reviewed and refined.

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

# APPENDIX

## A  Theoretical Details

**Defining Amortized Bayesian Inference**   The term "amortized" has been used inconsistently throughout the literature, often denoting different generalization scopes. To clarify this concept for the discussion within this work, we offer the following definition:

**Definition 1.** *Let $\mathcal{A}$ denote a learner, $\boldsymbol{y}$ denote target variables, $\boldsymbol{x}$ represent input data, and $\boldsymbol{c}$ denote context variables. A learner $\boldsymbol{y} \sim \mathcal{A}(\boldsymbol{x}, \boldsymbol{c})$ is an* amortized Bayesian approximator *of a target quantity $\boldsymbol{y}$ with respect to a joint distribution $p(\boldsymbol{x}, \boldsymbol{y}, \boldsymbol{c})$ if it can directly approximate $p(\boldsymbol{y} \mid \boldsymbol{x}, \boldsymbol{c})$ for any $(\boldsymbol{x}, \boldsymbol{c}) \sim p(\boldsymbol{x}, \boldsymbol{c})$ without requiring further training or additional approximation algorithms.*

By this definition, sequential methods that necessitate further training for new data (Papamakarios & Murray, 2016; Glöckler et al., 2022) are not considered amortized. Similarly, neural likelihood estimation (NLE; Papamakarios et al., 2019) and neural ratio estimation (NRE) (Hermans et al., 2020), which depend on MCMC algorithms, do not qualify as amortized. In contrast, recent transformer-based (Gloeckler et al., 2024) or context-aware methods (Elsemüller et al., 2024) clearly fall within the scope of amortized neural posterior estimation (NPE).

## B  Experimental Details

Since the analytic posterior is only obtainable in Experiment 2, we measure performance relative to the data-generating parameters $\boldsymbol{\theta}^*$ to enable a direct comparison between the experiments. For likelihood misspecification settings, $\boldsymbol{\theta}^*$ is closely related to the posterior $p(\boldsymbol{\theta} \mid \widetilde{\boldsymbol{x}}_{\mathrm{obs}})$ based on adjusted (e.g., decontaminated) data $\widetilde{\boldsymbol{x}}_{\mathrm{obs}}$ (**Target 2**). Thus, the NPE-UDA posterior approximations being closer to $\boldsymbol{\theta}^*$ than the analytic posterior $p(\boldsymbol{\theta} \mid \boldsymbol{x})$ in the contamination scenario of Experiment 2 indicates that NPE-UDA methods indeed focus **Target 2**.

In all experiments, we build upon the `BayesFlow` Python library for amortized Bayesian workflows using generative neural networks (Radev et al., 2023).

### B.1  Method Details

**NNPE**   We implemented NNPE following the original implementation of Ward et al. (2022) at `https://github.com/danielward27/rnpe`, who used a spike scale of $\sigma = 0.01$ and a slab scale of $\tau = 0.25$ for all experiments. To remain consistent with the original implementation of Ward et al. (2022), we applied NNPE to standardized data in all experiments (in Experiment 3, we applied equivalent scaling instead, see Appendix B.5). Whether spike (standard normal) or slab (standard Cauchy) noise is applied to a simulated data point is determined by sampling from a Bernoulli distribution with $p = 0.5$.

**Sensitivities of NPE-UDA**   In both experiments, we found the typical UDA phenomenon of sensitivity to higher learning rates (Perone et al., 2019) in the form of unstable learning dynamics such as exploding gradients. We also found sensitivity to short training times, suggesting that finding a stable optimum for the two-component NPE-UDA loss in Eq. 3 requires more gradient updates than usual.

**Computational Cost of NPE-UDA**   NPE-UDA methods only modify the training process of standard NPE and therefore do not influence the computational cost of inference. Since NPE-UDA methods operate in the compressed summary space, the runtime increase during training is minimal compared to NPE. Experiment 3 features the largest (32-dimensional) summary space of our experiments and thus constitutes an upper bound measurement of the time increase due to integrating UDA: Here, NPE and NPE-MMD took 12s/epoch (amounting to a total wall clock time of $\sim$ 4 minutes) and NPE-DANN 13s/epoch (amounting to a total wall clock time of $\sim$ 4 minutes and 20 seconds) during GPU training on a cluster.

### B.2 Metrics

We compute multiple metrics that measure the performance based on the approximation performance of $J$ data-generating parameters $\{\theta_j^*\}_{j=1}^J$ via $S$ posterior samples (we forego the obs notation where possible for brevity here). Depending on the metric, results are averaged across the $J$ parameters and/or $N$ observed data sets.

#### B.2.1 Parameter Space Performance Metrics

**Negative log likelihood (NLL):**

$$\text{NLL} = -\frac{1}{N} \sum_{n=1}^{N} \log q(\boldsymbol{\theta}_n^* \mid \phi(\boldsymbol{x}_n)), \tag{8}$$

where we utilize the fact that normalizing flows allow us to easily compute approximate (log) densities.

**Normalized root mean squared error (NRMSE):**

$$\text{NRMSE} = \frac{1}{N} \sum_{n=1}^{N} \left[ \frac{1}{J} \sum_{j=1}^{J} \frac{\sqrt{\frac{1}{S} \sum_{s=1}^{S} (\theta_{j,n}^* - \hat{\theta}_{j,n}^{(s)})^2}}{\max(\theta_j^*) - \min(\theta_j^*)} \right]. \tag{9}$$

**Expected calibration error (ECE)** via the fraction of ground-truth inliers for $R$ linearly spaced $\alpha$-confidence intervals in $[0.005, 0.995]$ (Ardizzone et al., 2018; Radev et al., 2020):

$$\text{ECE} = \frac{1}{J} \sum_{j=1}^{J} \text{median}_{r=1}^{R} \left( \left| \frac{1}{N} \sum_{n=1}^{N} \mathbb{I}\left\{ Q_{\frac{1-\alpha_r}{2}}(\hat{\theta}_j^{(n)}) \le \theta_j^* \le Q_{1-\frac{1-\alpha_r}{2}}(\hat{\theta}_j^{(n)}) \right\} - \alpha_r \right| \right), \tag{10}$$

where $\text{median}_{r=1}^{R}$ represents the median fraction of inliers across the $R = 20$ credible intervals and $Q_k(\hat{\theta}_j^{(n)})$ represents the $k$-th quantile of the posterior samples for the $n$-th data set. We estimate the ECE on all test data sets via the median calibration error of $R = 20$ linearly spaced credible intervals, averaged across $J$ model parameters.

**Posterior contraction (PC)** relative to the prior distribution (Betancourt, 2018):

$$\text{PC} = \frac{1}{N} \sum_{n=1}^{N} \left[ \frac{1}{J} \sum_{j=1}^{J} \left( 1 - \frac{\text{Var}(\hat{\theta}_{j,n}^{(s)})}{\text{Var}(\theta_{j,n}^*)} \right) \right]. \tag{11}$$

#### B.2.2 Data Space Performance Metrics

**Posterior predictive distance (PPD):**

$$\text{PPD} = \frac{1}{N} \sum_{n=1}^{N} \left[ \frac{1}{S} \sum_{s=1}^{S} d\left( \boldsymbol{x}_n, \hat{\boldsymbol{x}}_n^{(s)} \right) \right]. \tag{12}$$

where $\hat{\boldsymbol{x}}^{(s)}$ represents a re-simulation based on a posterior sample of all estimated parameters, $\hat{\boldsymbol{\theta}}^{(s)}$, and we use RMSE for $d(\cdot, \cdot)$. To keep computation times reasonable, we limit the number of re-simulations $\hat{\boldsymbol{x}}^{(s)}$ by using a random subset of all $S$ posterior samples in the experiments with a large number of posterior samples (Experiment 1: 100 samples; Experiment 2: 100 samples). We investigate two different PPD variants: (i) the default calculation defined in Equation 12 with $\boldsymbol{x}_n$ as the reference data set, (ii) and a calculation where $\boldsymbol{x}_n$ is replaced with a "denoised" reference data set $\widetilde{\boldsymbol{x}}_n$ that is obtained by re-simulating from the ground-truth parameter in the synthetic experiments and intensive data pre-processing in the real-world experiment. One limitation of the PPD metric is that any parameter values that would break the simulator have to be excluded before re-simulating, which can reduce the sensitivity of the metric for detecting approximation failures.

### B.2.3 Network Space Metrics

**Summary space domain distance (SSDD)**: SSDD, which does not quantify approximation performance but the degree of summary space alignment, is measured with two variants. The first variant is based on the biased sample-based $\widehat{\mathrm{MMD}}^2$ estimator (Gretton et al., 2012):

$$\mathrm{SSDD}_{\mathrm{MMD}} = \frac{1}{N} \sum_{n=1}^{N} \widehat{\mathrm{MMD}}^2 \big[ \{\phi(\boldsymbol{x}_n)\} \, \| \, \{\phi(\boldsymbol{x}_n^{\mathrm{obs}})\} \big], \tag{13}$$

where $\{\phi(\boldsymbol{x}_n)\}$ and $\{\phi(\boldsymbol{x}_n^{\mathrm{obs}})\}$ are sets of summary statistics over which the expectations are approximated. The second variant $\mathrm{SSDD}_{\mathrm{C2ST}}$ is based on the classifier two-sample test (C2ST) and represents the accuracy of an MLP classifier trained to distinguish the sets of summary statistics $\{\phi(\boldsymbol{x}_n)\}$ and $\{\phi(\boldsymbol{x}_n^{\mathrm{obs}})\}$ (Bischoff et al., 2024).

**Inference network latent distance (INLD)**: Following (Siahkoohi et al., 2023), we consider distortions in the inference network's latent space a proxy for approximation quality, which is a direct consequence of maximum likelihood training. To measure general distortions (beyond location and scale), we again consider the biased sample-based $\widehat{\mathrm{MMD}}^2$ estimator (Gretton et al., 2012):

$$\mathrm{INLD}_{\mathrm{MMD}} = \frac{1}{N} \sum_{n=1}^{N} \widehat{\mathrm{MMD}}^2 \big[ \{\boldsymbol{z}_n\} \, \| \, \{f(\boldsymbol{\theta}_n; \boldsymbol{x}_n)\} \big], \tag{14}$$

where $f(\boldsymbol{\theta}_n; \boldsymbol{x}_n)$ denotes the forward direction of the conditional invertible network realizing the normalizing flow.

## B.3 Experiment 1 - Ricker

**Probabilistic Model**   We follow the model specification of Radev et al. (2020), including their prior specifications for the growth rate parameter $r$ and the scaling parameter $\rho$ (see Radev et al. (2020) for details). The only deviation from Radev et al. (2020) is the specification of the parameter $\sigma$ governing the standard deviation of Gaussian noise, where we follow Huang et al. (2023) and fix $\sigma = 0.3$ to allow for an easy visual inspection of the resulting 2D posterior landscape during method development.

**Network Architecture**   We use an LSTNet architecture as described in Zhang & Mikelsons (2023) for the summary network $\phi$, compressing the input to 6-dimensional summary statistics. For the generative inference network $q$, we use an affine coupling flow architecture (Ardizzone et al., 2021; Kingma & Dhariwal, 2018) with 3 coupling layers. See Table B.1 for the optimized hyperparameters (regarding architectural as well as training choices) per method and their respective search range.

Table B.1: Hyperparameter search ranges for all methods.

| Hyperparameter | Range | Method(s) |
|---|---|---|
| Initial learning rate ($\alpha$) | $1 \times 10^{-4}$–$5 \times 10^{-3}$ | NPE, NNPE, NPE-MMD, NPE-DANN |
| NPE-UDA alignment weight $\lambda$ | 0.01–150.0 | NPE-MMD, NPE-DANN |
| Discriminator depth | 2–4 | NPE-DANN |
| Discriminator width | 128–1024 | NPE-DANN |
| Gradient reversal weight $\lambda_{grl}$ | 0.5–15.0 | NPE-DANN |
| Label smoothing | 0.0–0.3 | NPE-DANN |

**Training and Evaluation Details**   We use an AdamW optimizer with an initial learning rate set by the hyperparameter search algorithm and cosine decay. We further use a batch size of 32 and train for 20 epochs. For the evaluation, we generate $S = 5\,000$ posterior samples per method and test data set.

**Additional Results**  Figure B1 contains all performance metrics for the additional NPE-DANN hyperparameters, showing an overall little effect of the additional hyperparameters. We therefore favor simple settings of these hyperparameters in the following experiments.

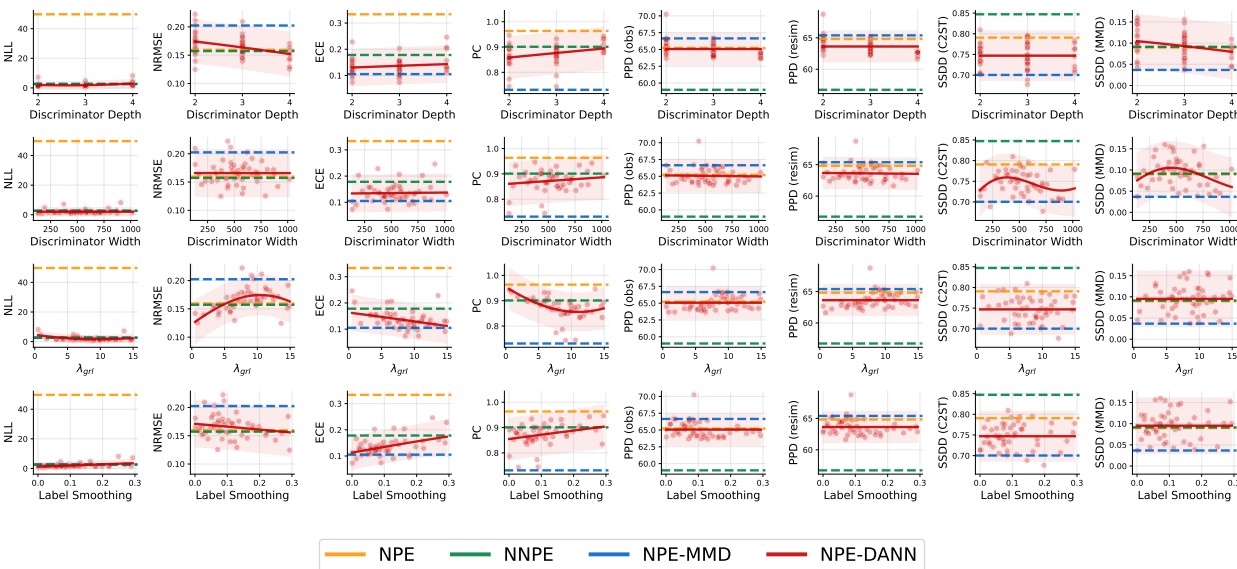

Figure B1: **Experiment 1**. Performance metrics for the additional NPE-DANN hyperparameters resulting from 50 separate Bayesian hyperparameter optimization runs per method. The solid trend lines represent the predictive mean of a Gaussian process regression fitted to the individual run results, with the shaded areas representing 95% confidence intervals of the predictive distribution. If a parameter was not optimized, the method's average performance is depicted by a dashed horizontal line. Lower values indicate better performance for all metrics but PC. NLL = Negative Log Likelihood. NRMSE = Normalized Root Mean Squared Error. ECE = Expected Calibration Error. PC = Posterior Contraction.

## B.4  Experiment 2 - 2D Gaussian Means

| Misspecification Setting | Prior | Likelihood |
|---|---|---|
| Well-specified | $\boldsymbol{\mu} \sim \mathcal{N}(\mathbf{0}, \boldsymbol{I}_2)$ | $\boldsymbol{x}_k \sim \mathcal{N}(\boldsymbol{\mu}, \boldsymbol{I}_2)$ |
| Prior location misspecification | $\boldsymbol{\mu} \sim \mathcal{N}(\boldsymbol{\mu_0}, \boldsymbol{I}_2)$ | $\boldsymbol{x}_k \sim \mathcal{N}(\boldsymbol{\mu}, \boldsymbol{I}_2)$ |
| Prior scale misspecification | $\boldsymbol{\mu} \sim \mathcal{N}(\mathbf{0}, \tau_0\boldsymbol{I}_2)$ | $\boldsymbol{x}_k \sim \mathcal{N}(\boldsymbol{\mu}, \boldsymbol{I}_2)$ |
| Likelihood scale misspecification | $\boldsymbol{\mu} \sim \mathcal{N}(\mathbf{0}, \boldsymbol{I}_2)$ | $\boldsymbol{x}_k \sim \mathcal{N}(\boldsymbol{\mu}, \tau\boldsymbol{I}_2)$ |
| Contamination misspecification | $\boldsymbol{\mu} \sim \mathcal{N}(\mathbf{0}, \boldsymbol{I}_2)$ | $\boldsymbol{x}_k \sim \frac{\epsilon}{2} \cdot \delta(\boldsymbol{x} - \boldsymbol{c}) + \frac{\epsilon}{2} \cdot \delta(\boldsymbol{x} + \boldsymbol{c}) + (1 - \epsilon) \cdot \mathcal{N}(\mu, \boldsymbol{I}_2)$ |

Table B.2: **Experiment 2**: Overview of the model specifications in the different misspecification settings.

Table B.2 provides an overview of the well-specified setting and the different misspecification scenarios inspired by Schmitt et al. (2024).

**Network Architecture**  We use a deep set architecture (Zaheer et al., 2017) for the summary network $\phi$, compressing the input to 4-dimensional summary statistics. For the generative inference network $q$, we use an affine coupling flow architecture (Ardizzone et al., 2021; Kingma & Dhariwal, 2018) with 3 coupling layers.

For the domain classifier $\psi$ in NPE-DANN, we use a standard feedforward network with 2 hidden layers of width 256. We do not use label smoothing or weight the gradient reversal balance.

**Training and Evaluation Details** To rule out any overfitting effects on the results, we use an online training approach where new data from the simulated and the observed domain is simulated at each training step, resulting in overall simulation budgets of $N = 48\,000$ and $N_{\mathrm{obs}} = 49\,000$. Since we use a batch size of 32, also for the observed data in NPE-UDA methods, online training amounts to $1\,500$ mini-batches and thus gradient updates. We use an Adam optimizer with an initial learning rate of $5 \cdot 10^{-4}$ and cosine decay. We generate $S = 5\,000$ posterior samples per method and test data set.

We provide additional results iterating over (i) performance in the simulated vs. the observed domain and (ii) $\lambda = [0.1, 1, 10]$.

**Additional Results** Figure B2, Figure B3, and Figure B4 show the performance in the simulated domain for varying $\lambda$ settings. Despite notable performance differences in the observed domain, all methods perform well in the simulated domain for the vast majority of settings, with the only exception being failures of NPE-UDA methods for high regularization weights in Figure B4. NNPE performs worse in the simulated (noiseless) domain since it was optimized based on noisy training data. Besides the clear NPE-UDA failures in $\lambda = 10$ settings, we mostly do not observe a trade-off of the summary space alignment of the NPE-UDA methods in the simulated domain.

Figure B5 and Figure B6 show the performance in the observed domain for varying $\lambda$ settings. The results confirm our finding of an application- and also method-specific $\lambda$ optimum: Whereas the difference of the NPE-UDA methods to NPE is often small for $\lambda = 0.1$, $\lambda = 10$ renders the NPE-UDA methods highly unstable under misspecification, especially when large domain shifts are present.

Figure B7 directly contrasts the effects of prior misspecification due to too narrowly chosen priors (see "Prior Scale" results in Figure 5) with the complimentary scenario of too broadly chosen priors. While too narrow priors that exclude relevant parts of the prior space constitute a clear misspecification, the case is less clear for too broad priors, to which practitioners commonly lean when in doubt. In Figure B7, we indeed find that, for this experimental setting, too broad priors are less detrimental for NPE than too narrow priors. The symmetric summary space domain distance (SSDD) of NPE (partially overlapping with NNPE in the plot) when deviating from the well-specified setting suggests that both regimes are equally atypical from the perspective of the trained network. However, while we observe drastic differences between the analytic posterior and the NPE methods in terms of parameter recovery error (NRMSE), posterior predictive data space distance (PPD), and calibration (ECE) in the too-narrow-priors regime (right side), these errors are much less pronounced in the too-broad-priors regime (left side). Therefore, the differences between the various NPE methods in the too-broad-priors regime are mainly constrained to different contraction (1 - PC) vs. calibration (ECE) trade-offs. Here, the contraction patterns mirror the general trend of NPE-UDA posteriors across our experiment to reduce the influence of unmodeled information in the observed data and thus stay closer to the prior than NPE (i.e., less contraction equals broader NPE-UDA posteriors in the too-broad-priors regime and vice versa). We hypothesize that the dimensionality of the inference problem plays a crucial role: In this low-dimensional (2D) inference task, the too-broad-priors regime is a much less clear misspecification/out-of-distribution setting than the too-narrow-priors regime, since in the former regime, the test setting is still well-represented in the typical set of the simulated training data. As dimensionality increases, we expect this to become less and less the case due to the curse of dimensionality. Thus, for high-dimensional inference problems, we expect the choice of overly broad priors to become as detrimental as overly narrow priors. Comprehensive results for the too-broad-priors regime for all $\lambda$ settings are displayed in Figure B8 for the simulated (well-specified) domain and Figure B9 for the observed domain.

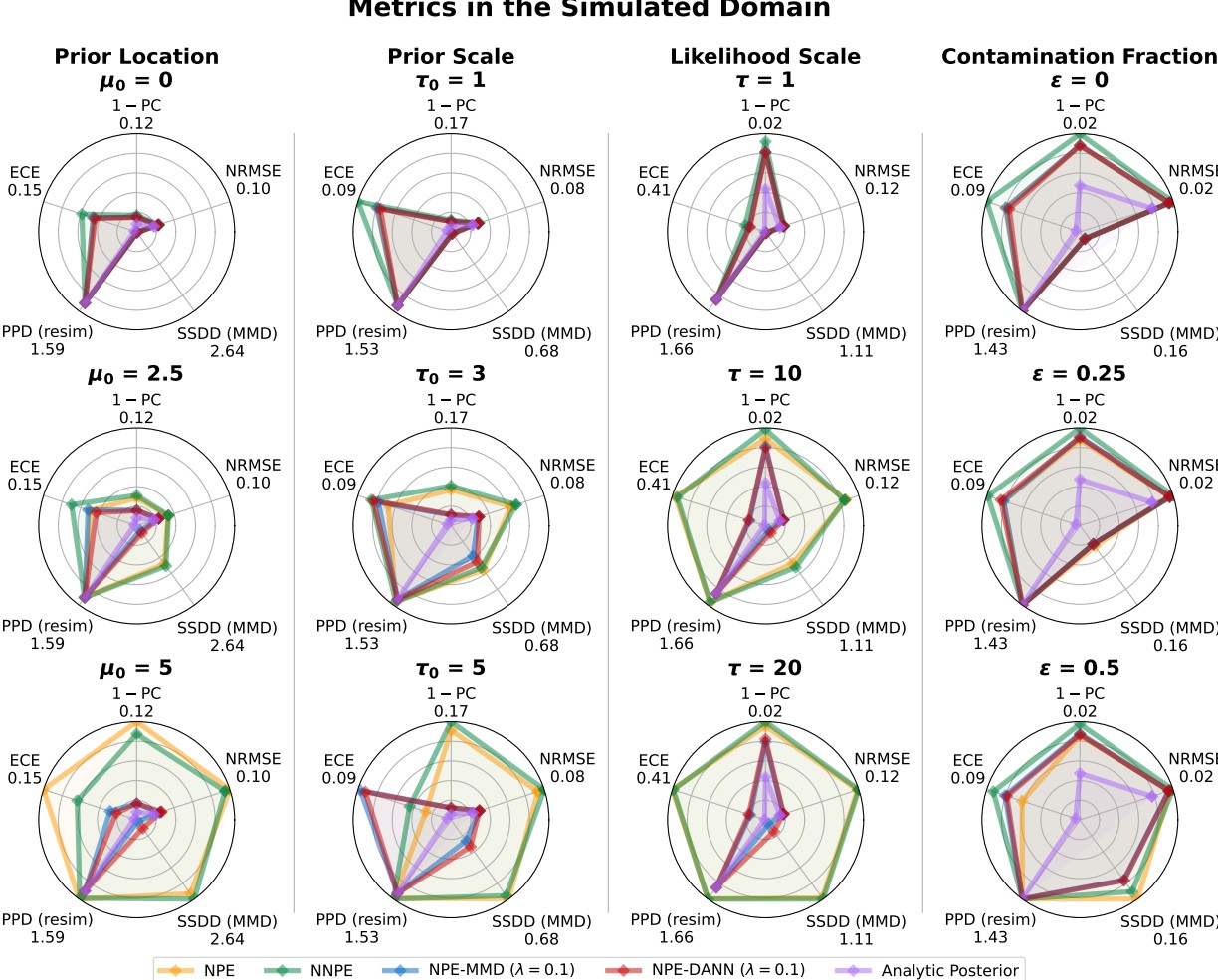

Figure B2: **Experiment 2**. Performance metrics and summary space domain distance (SSDD) of the methods in all misspecification scenarios (columns) **on simulated (i.e., well-specified) data for $\lambda = 0.1$ in NPE-MMD and NPE-DANN**, aggregated via the median of 10 runs. Lower values indicate better performance for all metrics but SSDD. $1-$ PC $= 1-$ Posterior Contraction. NRMSE $=$ Normalized Root Mean Squared Error. SSDD (MMD) $=$ Summary Space Domain Distance measured via MMD (not applicable for Analytic Posterior). PPD (resim) $=$ Posterior Predictive Distance measured via the RMSE to resimulated data. ECE $=$ Expected Calibration Error.

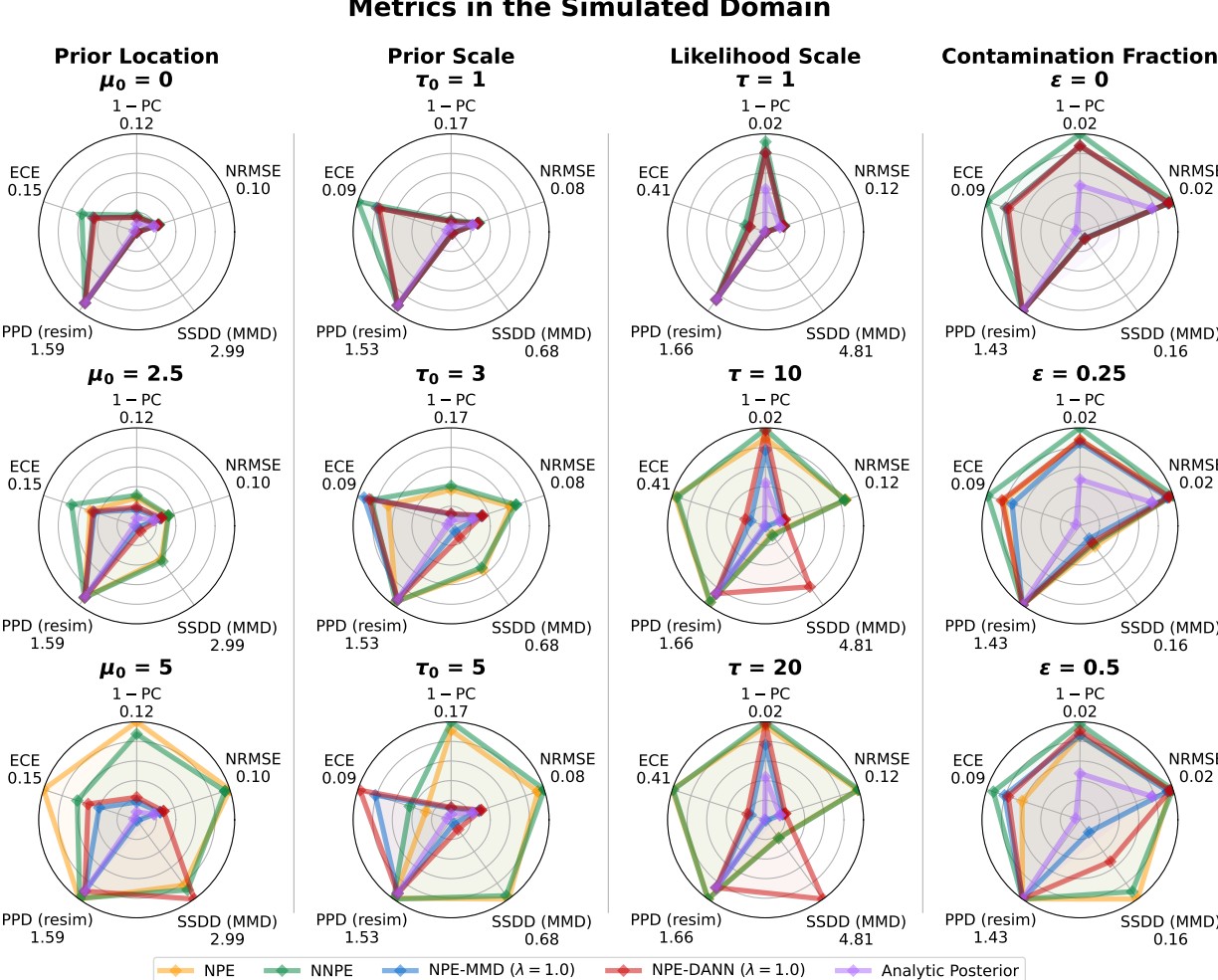

Figure B3: **Experiment 2**. Performance metrics and summary space domain distance (SSDD) of the methods in all misspecification scenarios (columns) **on simulated (i.e., well-specified) data for $\lambda = 1$ in NPE-MMD and NPE-DANN**, aggregated via the median of 10 runs. Lower values indicate better performance for all metrics but SSDD. $1-$ PC $= 1-$ Posterior Contraction. NRMSE = Normalized Root Mean Squared Error. SSDD (MMD) = Summary Space Domain Distance measured via MMD (not applicable for Analytic Posterior). PPD (resim) = Posterior Predictive Distance measured via the RMSE to resimulated data. ECE = Expected Calibration Error.

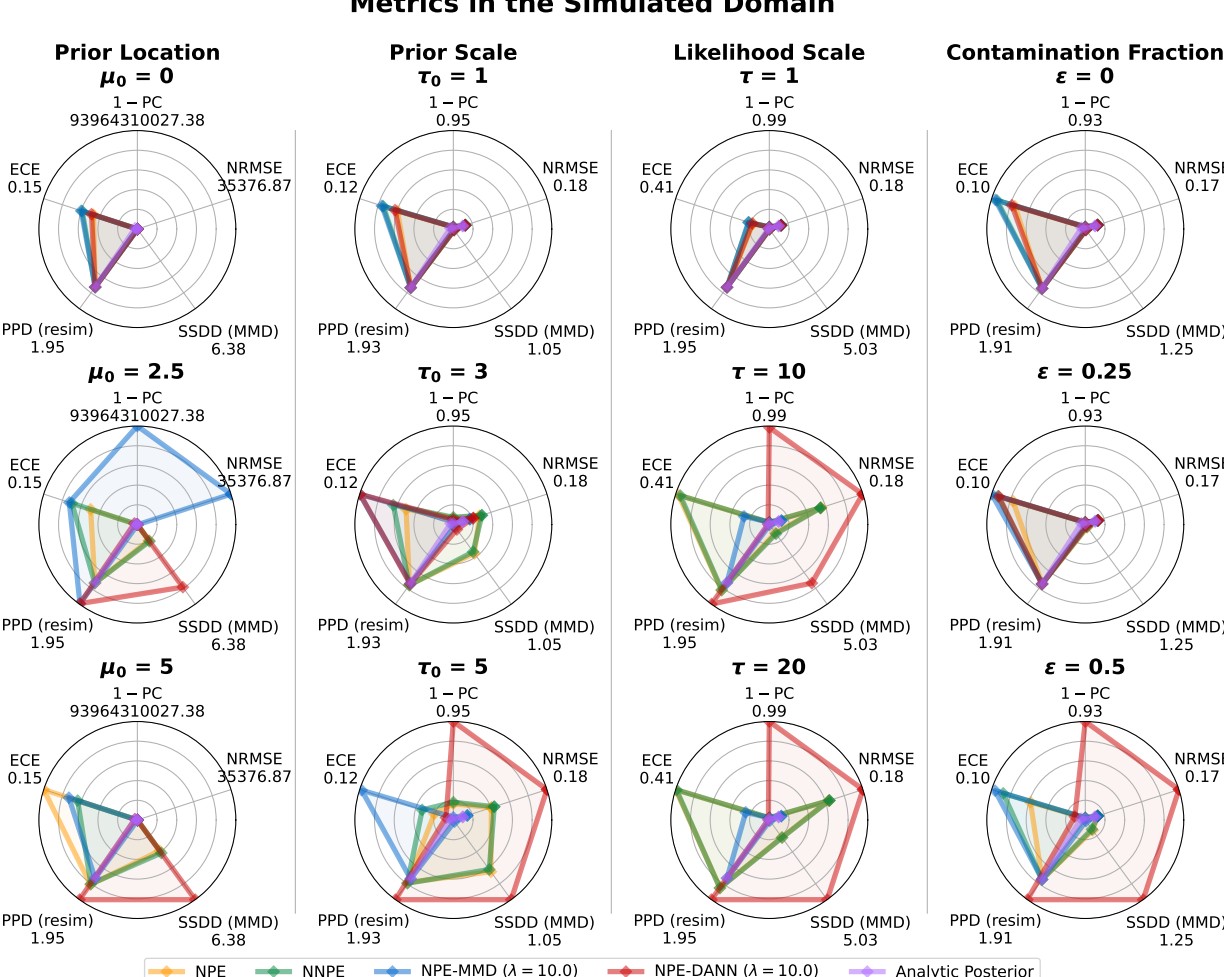

Figure B4: **Experiment 2**. Performance metrics and summary space domain distance (SSDD) of the methods in all misspecification scenarios (columns) **on simulated (i.e., well-specified) data for $\lambda = 10$ in NPE-MMD and NPE-DANN**, aggregated via the median of 10 runs. Lower values indicate better performance for all metrics but SSDD. $1-$ PC $= 1-$ Posterior Contraction. NRMSE = Normalized Root Mean Squared Error. SSDD (MMD) = Summary Space Domain Distance measured via MMD (not applicable for Analytic Posterior). PPD (resim) = Posterior Predictive Distance measured via the RMSE to resimulated data. ECE = Expected Calibration Error.

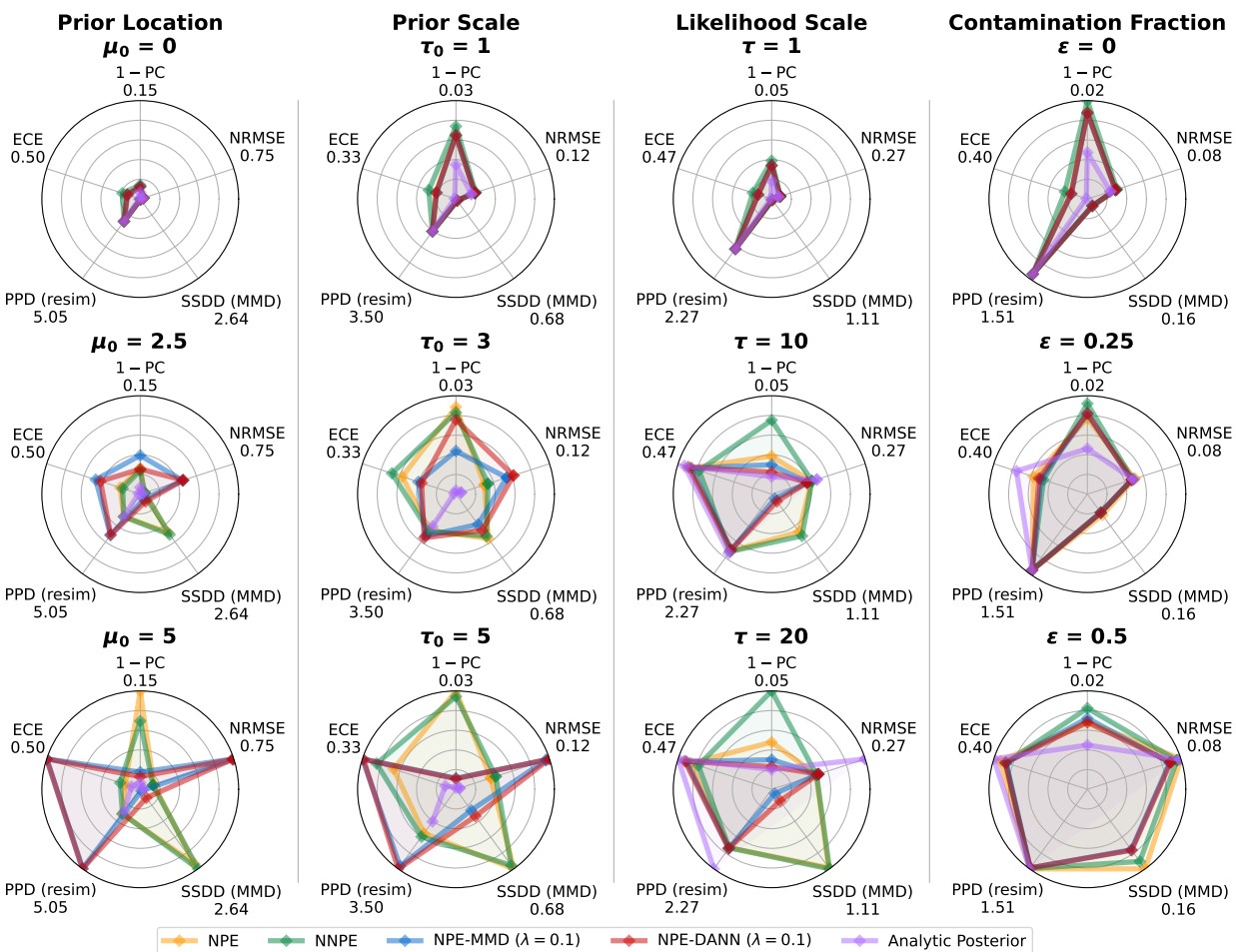

Figure B5: **Experiment 2**. Performance metrics and summary space domain distance (SSDD) of the methods in all misspecification scenarios (columns) **for $\lambda = 0.1$ in NPE-MMD and NPE-DANN**, aggregated via the median of 10 runs. Lower values indicate better performance for all metrics but SSDD. $1-$ PC $= 1-$ Posterior Contraction. NRMSE = Normalized Root Mean Squared Error. SSDD (MMD) = Summary Space Domain Distance measured via MMD (not applicable for Analytic Posterior). PPD (resim) = Posterior Predictive Distance measured via the RMSE to resimulated data. ECE = Expected Calibration Error.

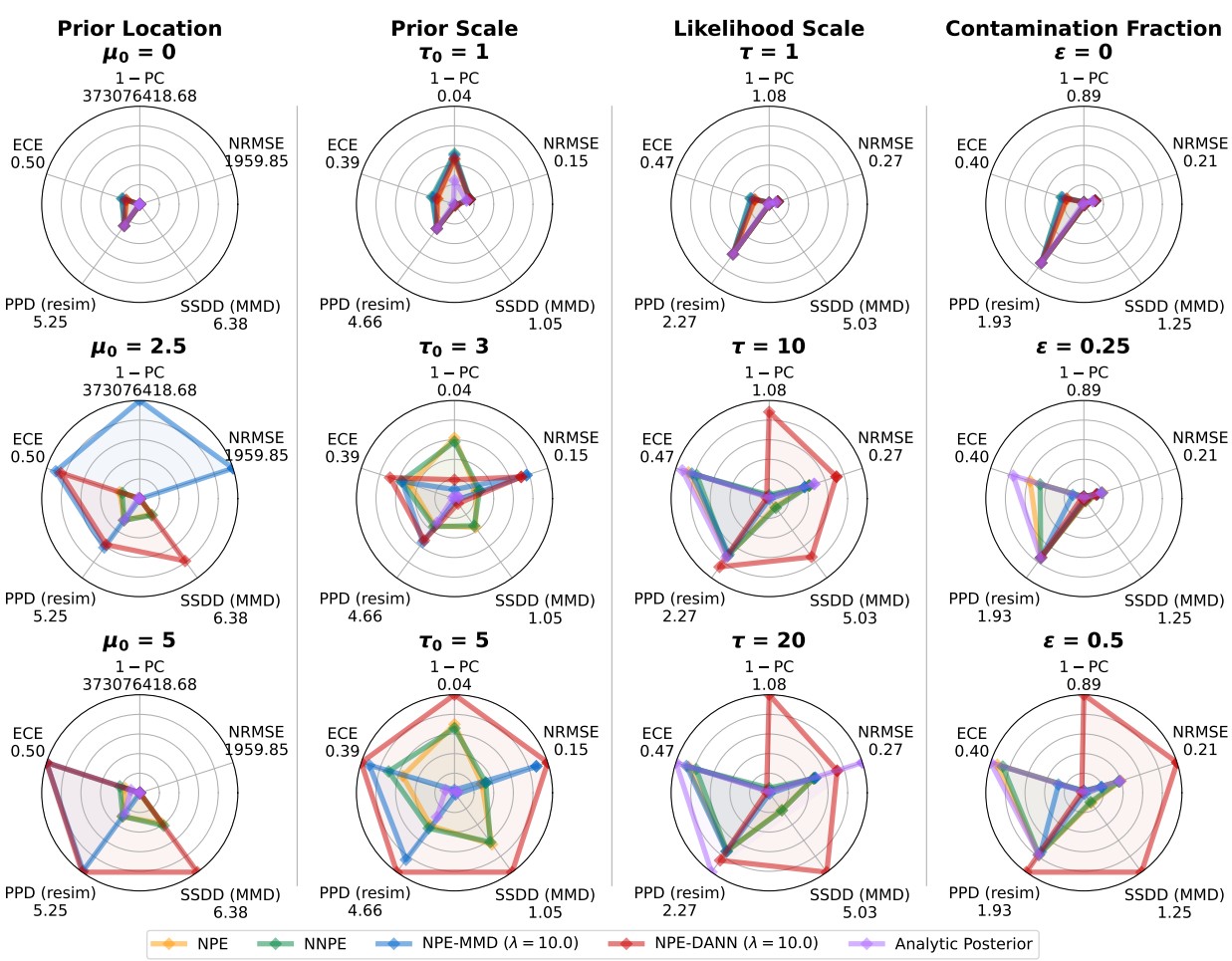

Figure B6: **Experiment 2**. Performance metrics and summary space domain distance (SSDD) of the methods in all misspecification scenarios (columns) **for $\lambda = 10$ in NPE-MMD and NPE-DANN**, aggregated via the median of 10 runs. Lower values indicate better performance for all metrics but SSDD. $1-$ PC = $1-$ Posterior Contraction. NRMSE = Normalized Root Mean Squared Error. SSDD (MMD) = Summary Space Domain Distance measured via MMD (not applicable for Analytic Posterior). PPD (resim) = Posterior Predictive Distance measured via the RMSE to resimulated data. ECE = Expected Calibration Error.

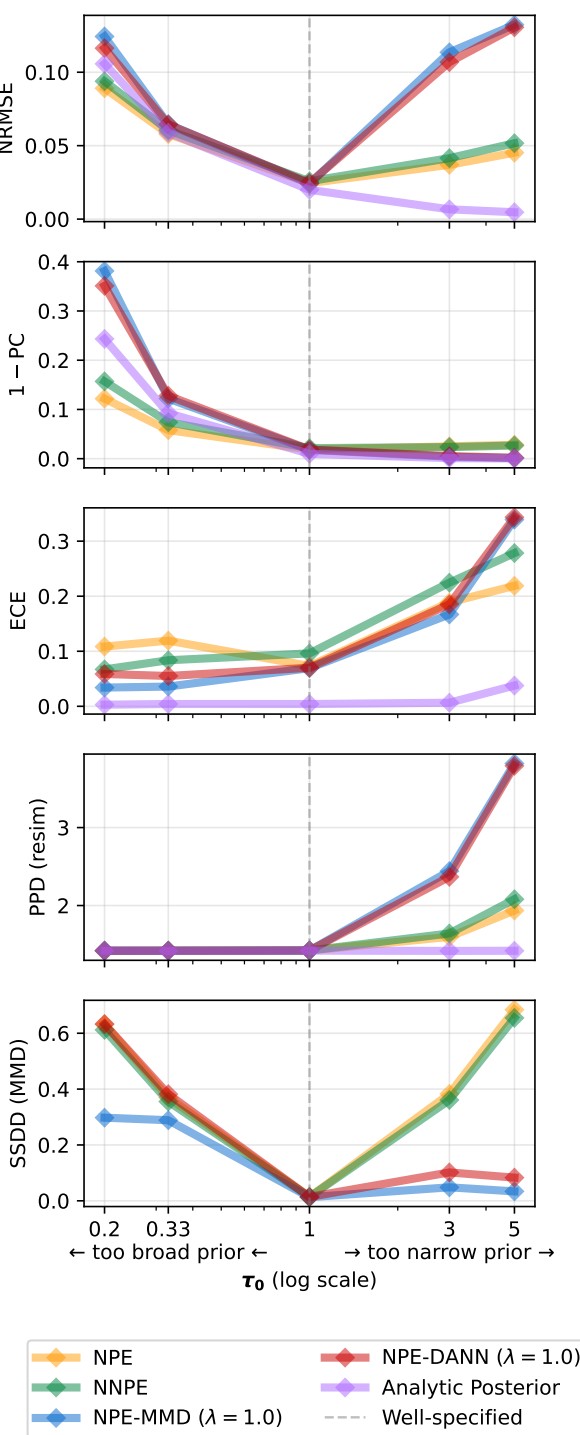

Figure B7: **Experiment 2**. Performance metrics and summary space domain distance (SSDD) of the methods in all misspecification scenarios (columns) **for prior scale misspecification with too broad vs. too narrow priors ($\lambda = 1$)**, aggregated via the median of 10 runs. Lower values indicate better performance for all metrics but SSDD. NRMSE = Normalized Root Mean Squared Error. $1-$ PC = $1-$ Posterior Contraction. ECE = Expected Calibration Error. PPD (resim) = Posterior Predictive Distance measured via the RMSE to resimulated data. SSDD (MMD) = Summary Space Domain Distance measured via MMD (not applicable for Analytic Posterior).

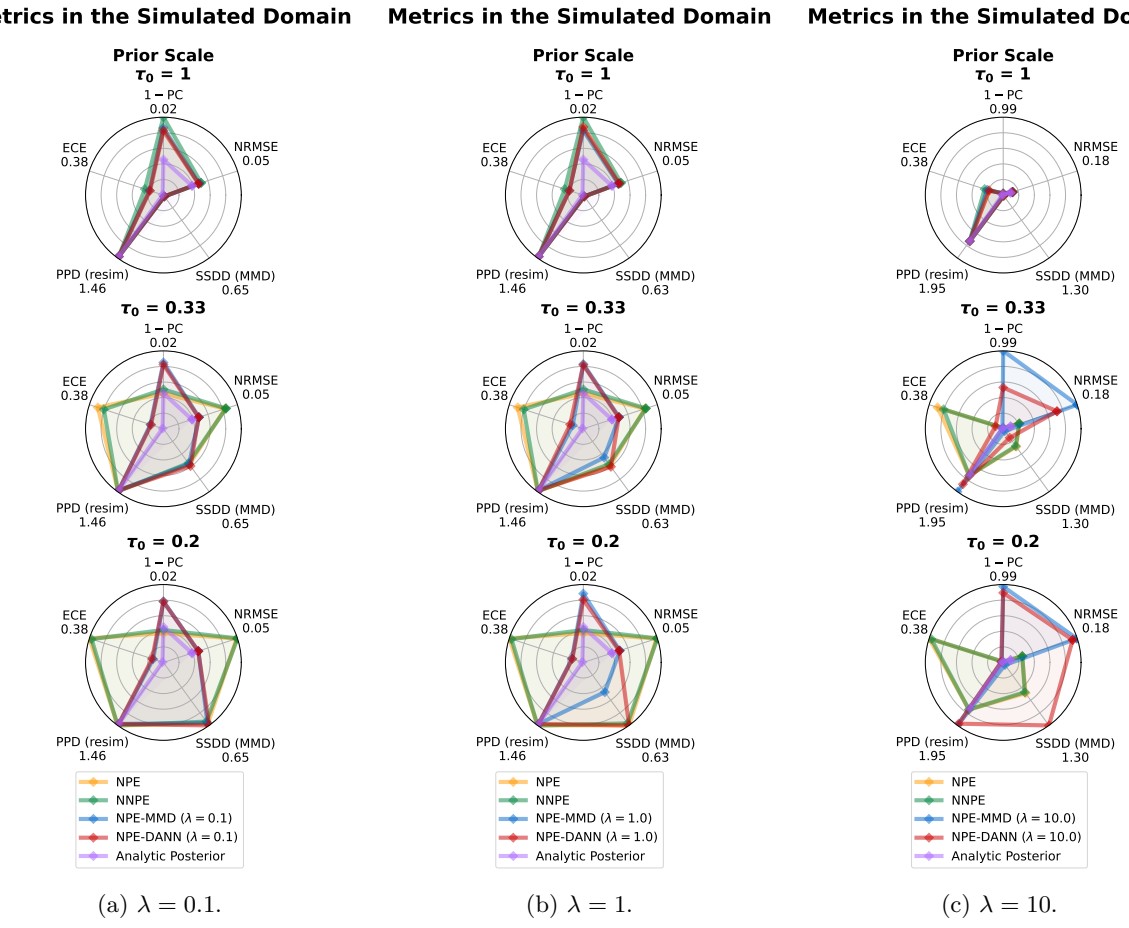

Figure B8: **Experiment 2**. Performance metrics and summary space domain distance (SSDD) of the methods in all misspecification scenarios (columns) **on simulated (i.e., well-specified) data for prior scale misspecification with too broad priors**, aggregated via the median of 10 runs. Lower values indicate better performance for all metrics but SSDD. $1-$ PC $= 1-$ Posterior Contraction. NRMSE = Normalized Root Mean Squared Error. SSDD (MMD) = Summary Space Domain Distance measured via MMD (not applicable for Analytic Posterior). PPD (resim) = Posterior Predictive Distance measured via the RMSE to resimulated data. ECE = Expected Calibration Error.

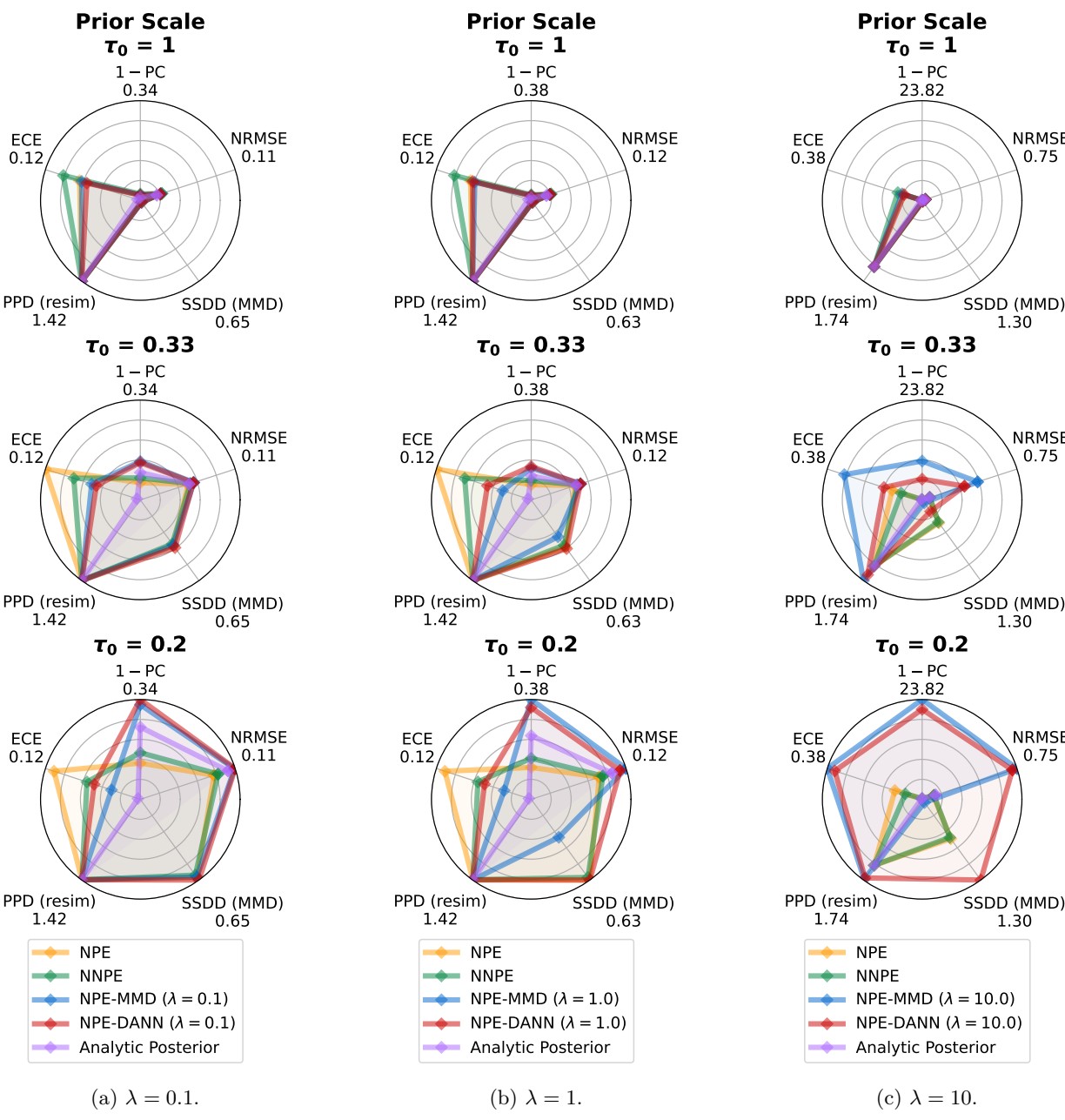

Figure B9: **Experiment 2**. Performance metrics and summary space domain distance (SSDD) of the methods in all misspecification scenarios (columns) **for prior scale misspecification with too broad priors**, aggregated via the median of 10 runs. Lower values indicate better performance for all metrics but SSDD. $1- PC = 1-$ Posterior Contraction. NRMSE = Normalized Root Mean Squared Error. SSDD (MMD) = Summary Space Domain Distance measured via MMD (not applicable for Analytic Posterior). PPD (resim) = Posterior Predictive Distance measured via the RMSE to resimulated data. ECE = Expected Calibration Error.

### B.5 Experiment 3

**Probabilistic Model**   We adopt a noisy camera model similar to the one presented in Ramesh et al. (2022). First, the input image is clipped to the range $[-1, 1]$. Next, we use scikit-image (van der Walt et al., 2014) to add Poisson noise to the image, then filter it using a Gaussian filter from SciPy (Virtanen et al., 2020) with a standard deviation $\sigma$ for the Gaussian kernel. The result is a blurred image with identical size as the input image.

**Data Preparation**   For each data set, we normalize the images to the range $[-1, 1]$. The MNIST (Lecun et al., 1998) images are rescaled from $28 \times 28$ to $16 \times 16$ with anti-aliasing enabled. To produce the training data $\boldsymbol{x}$, the images are processed by the simulator, with $\sigma_0 = 1.4$. We adapt NNPE, which is originally defined for standardized data, by scaling the noise scales by the standard deviation $\sigma_{\boldsymbol{x}}$ of the training data.

While the training data remains constant across scenarios, the observed data is generated in different ways. For the prior misspecification scenario, we use the USPS data set (Hull, 1994) instead of MNIST, but the parameters of the simulator remain identical (i.e., $\sigma = \sigma_0$). For the likelihood scale scenario, we use $\widetilde{\sigma} = 1.25 \cdot \sigma_0$, leading to an increased blur. For the noise contamination scenario, we randomly set 10% of the pixels of each observation to black or white. For the row contamination scenario, we randomly set 2 rows of each observation (i.e., 12.5% of the pixels) to black. Refer to Table 3 for samples from each scenario.

**Network Architecture**   For the summary network, we use a 4-layer convolutional neural network, which outputs 32 learned summary variables.

For the inference network, we use flow matching (Lipman et al., 2023; Wildberger et al., 2023) to convert a multivariate Gaussian distribution to the approximate posterior distribution. We use a U-Net architecture (Ronneberger et al., 2015) to learn the flow field conditional on the summary variables.

For NPE-DANN, we use a domain classifier $\psi$ consisting of a standard feedforward network with 3 hidden layers of width 256, a gradient reversal layer (GRL) weight of 1, and no label smoothing.

**Training and Evaluation Details**   We use an AdamW optimizer with an initial learning rate of $5 \cdot 10^{-4}$ and cosine decay. We use a batch size of 32 and train for 20 epochs, except for NPE-MMD, which required increasing the batch size to 128. To keep the number of gradient updates constant, we also increased the number of epochs to 80 for NPE-MMD. The training budget is 50 000 training images, and 1 000 observed images. Training one neural network takes approximately 10 minutes on a GPU.

We use a moderate number of posterior samples of $S = 100$ per method and test data set to limit the computational cost of the experiment, allowing for a broader exploration of hyperparameters and the variance between multiple runs.

**Additional Results**   Table B.3 displays the performance on a held-out in-distribution data set, to assess the influence on the loss on the in-domain observations. We consistently observe similar performance for NPE and the NPE-UDA methods for low $\lambda$ and the expected performance loss of the noisy NNPE training on in-distribution data. Interestingly, NPE-MMD with low $\lambda$ shows a slight but consistent improvement over NPE in this setting. The reason for this is unclear—the additional observed data might serve as an extra training signal that improves the learned summary statistics, even for in-distribution data. Additionally, we can verify that for NPE-MMD with $\lambda = 1.0$ with vanishing SSDD, the summary space does not contain information, as also the in-distribution performance drops to a value we expect of unconditional generation. Table B.4 displays the same data as Table 2, but with uncertainty indicators (standard deviation). Figure B10 shows the plots corresponding to Figure 2 for the remaining three scenarios.

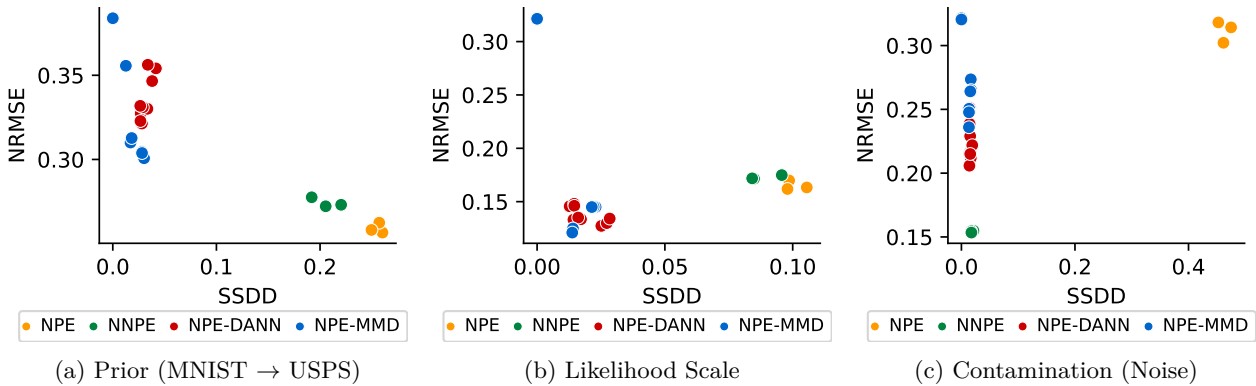

(a) Prior (MNIST → USPS)    (b) Likelihood Scale    (c) Contamination (Noise)

Figure B10: **Experiment 3**. Relationship of summary space domain distance (SSDD; MMD) and normalized root mean squared error (NRMSE, lower is better). For a), we see that despite the reduced SSDD, there is no gain in performance. For b) and c), we observe a sweet spot at a low SSDD value, before performance drops again when approaching zero. Refer to Table 2 for numerical values.

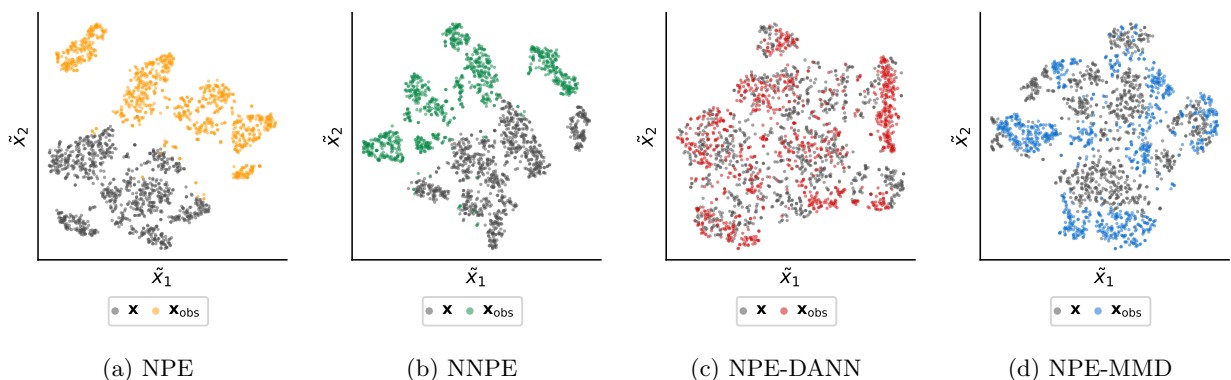

(a) NPE    (b) NNPE    (c) NPE-DANN    (d) NPE-MMD

Figure B11: **Experiment 3**. Prior: t-SNE representation of the summary spaces from the best run (lowest NRMSE) of each method. Please refer to Figure 6 for a detailed caption.

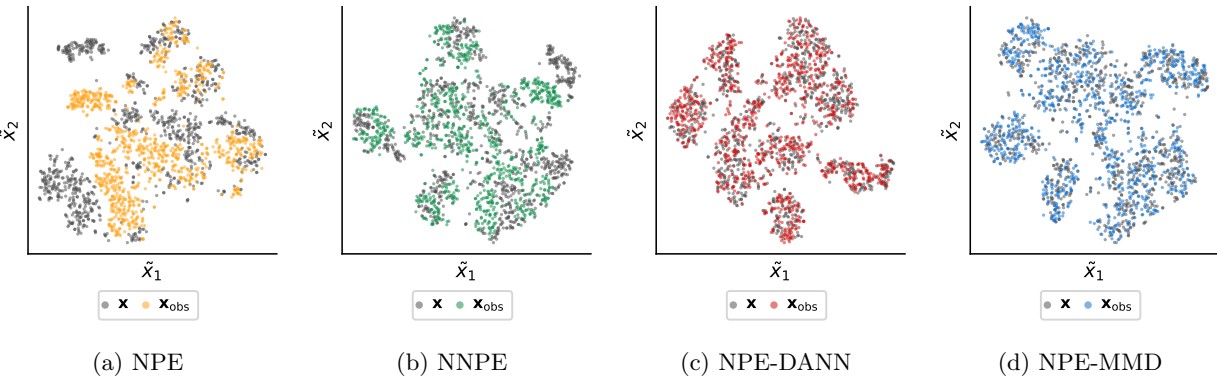

(a) NPE    (b) NNPE    (c) NPE-DANN    (d) NPE-MMD

Figure B12: **Experiment 3**. Likelihood (Scale): t-SNE representation of the summary spaces from the best run (lowest NRMSE) of each method. Please refer to Figure 6 for a detailed caption.

| Method | $\lambda$ | Prior (MNIST → USPS) | | Likelihood Scale | |
|---|---|---|---|---|---|
| | | NRMSE ↓ | PPD ↓ | NRMSE ↓ | PPD ↓ |
| NPE | - | 0.104 (1.6e-03) | 0.020 (3.0e-04) | 0.103 (9.9e-04) | 0.020 (2.0e-04) |
| NNPE | - | 0.140 (8.9e-04) | 0.030 (2.1e-04) | 0.141 (1.7e-04) | 0.031 (1.6e-04) |
| NPE-DANN | 0.01 | 0.116 (2.0e-03) | 0.024 (4.9e-04) | 0.110 (8.2e-04) | 0.023 (2.6e-04) |
| NPE-DANN | 0.10 | 0.163 (2.0e-02) | 0.037 (5.5e-03) | 0.122 (1.3e-03) | 0.026 (2.9e-04) |
| NPE-DANN | 1.00 | 0.267 (7.6e-03) | 0.080 (4.6e-03) | 0.140 (9.3e-04) | 0.030 (2.9e-04) |
| NPE-MMD | 0.01 | **0.092 (9.0e-04)** | 0.019 (2.7e-04) | **0.091 (3.2e-04)** | **0.018 (2.4e-04)** |
| NPE-MMD | 0.10 | 0.092 (1.8e-03) | **0.018 (2.7e-04)** | 0.169 (1.1e-01) | 0.055 (5.1e-02) |
| NPE-MMD | 1.00 | 0.298 (3.2e-02) | 0.111 (2.3e-02) | 0.320 (4.2e-04) | 0.127 (1.8e-04) |

| Method | $\lambda$ | Contamination (Noise) | | Contamination (Rows) | |
|---|---|---|---|---|---|
| | | NRMSE ↓ | PPD ↓ | NRMSE ↓ | PPD ↓ |
| NPE | - | 0.104 (1.8e-03) | 0.021 (2.7e-04) | 0.104 (1.3e-03) | 0.020 (2.7e-04) |
| NNPE | - | 0.141 (1.1e-03) | 0.031 (1.7e-04) | 0.139 (6.5e-04) | 0.030 (1.6e-04) |
| NPE-DANN | 0.01 | 0.114 (3.1e-04) | 0.023 (7.6e-05) | 0.115 (4.4e-03) | 0.023 (7.2e-04) |
| NPE-DANN | 0.10 | 0.139 (1.2e-03) | 0.029 (4.8e-04) | 0.136 (5.8e-03) | 0.028 (2.0e-03) |
| NPE-DANN | 1.00 | 0.192 (1.0e-02) | 0.045 (3.6e-03) | 0.179 (1.8e-02) | 0.040 (6.2e-03) |
| NPE-MMD | 0.01 | **0.091 (1.2e-03)** | 0.018 (2.3e-04) | **0.091 (5.9e-04)** | 0.018 (1.1e-04) |
| NPE-MMD | 0.10 | 0.092 (1.7e-03) | **0.018 (4.1e-04)** | 0.093 (6.8e-04) | **0.018 (9.4e-05)** |
| NPE-MMD | 1.00 | 0.319 (2.6e-04) | 0.128 (2.5e-04) | 0.320 (5.6e-04) | 0.128 (2.2e-04) |

Table B.3: **Experiment 3**. Overview of the metrics on a held-out validation data set from the training distribution (mean and standard deviation of three runs). NRMSE: Normalized Root Mean Squared Error (lower is better). PPD = Posterior Predictive Distance (RMSE) to resimulated data (lower is better). For NNPE and NPE-DANN, we see reduced performance on the training distribution. For NPE-MMD, we see that for successful runs, the performance on the training distribution improves. For settings with vanishing SSDD (compare Table 2), the performance drops massively, for both training distribution and observed distribution. This supports the notion that no meaningful information is learned in the summary space.

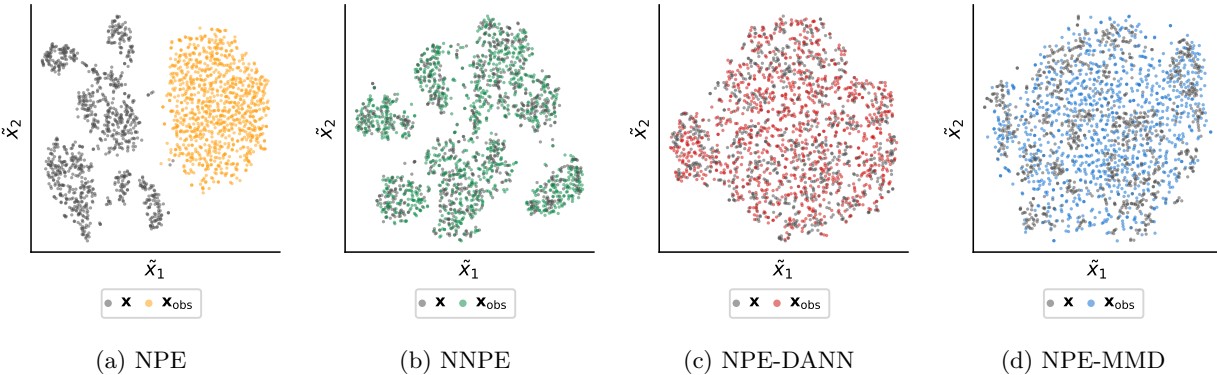

(a) NPE  (b) NNPE  (c) NPE-DANN  (d) NPE-MMD

Figure B13: **Experiment 3**. Contamination (Noise): t-SNE representation of the summary spaces from the best run (lowest NRMSE) of each method. Please refer to Figure 6 for a detailed caption.

| Method | $\lambda$ | Prior (MNIST → USPS) | | | Likelihood Scale | | |
|---|---|---|---|---|---|---|---|
| | | NRMSE ↓ | PPD ↓ | SSDD | NRMSE ↓ | PPD ↓ | SSDD |
| NPE | - | **0.259 (2.5e-03)** | **0.131 (1.7e-03)** | 0.256 (4.4e-03) | 0.165 (3.5e-03) | 0.036 (1.0e-03) | 0.101 (3.4e-03) |
| NNPE | - | 0.274 (2.3e-03) | 0.137 (1.6e-03) | 0.206 (1.2e-02) | 0.173 (1.5e-03) | 0.041 (1.3e-03) | 0.088 (5.3e-03) |
| NPE-DANN | 0.01 | 0.329 (4.1e-03) | 0.193 (2.6e-03) | 0.027 (1.1e-03) | **0.130 (2.8e-03)** | 0.031 (1.8e-03) | 0.027 (1.4e-03) |
| NPE-DANN | 0.10 | 0.326 (3.7e-03) | 0.193 (1.4e-03) | 0.029 (2.8e-03) | 0.134 (8.5e-04) | 0.031 (1.0e-03) | 0.016 (1.2e-03) |
| NPE-DANN | 1.00 | 0.352 (4.1e-03) | 0.205 (4.3e-03) | 0.038 (3.2e-03) | 0.147 (1.0e-03) | 0.032 (2.7e-04) | 0.014 (8.6e-04) |
| NPE-MMD | 0.01 | 0.303 (1.6e-03) | 0.184 (7.4e-04) | 0.029 (9.9e-04) | 0.145 (8.3e-05) | **0.027 (2.9e-04)** | 0.022 (6.1e-04) |
| NPE-MMD | 0.10 | 0.312 (1.1e-03) | 0.189 (5.7e-04) | 0.018 (4.6e-04) | 0.189 (9.4e-02) | 0.059 (4.9e-02) | 0.009 (6.5e-03) |
| NPE-MMD | 1.00 | 0.374 (1.3e-02) | 0.225 (1.1e-02) | 0.004 (5.9e-03) | 0.322 (2.1e-04) | 0.129 (2.7e-04) | 0.000 (2.4e-06) |

| Method | $\lambda$ | Contamination (Noise) | | | Contamination (Rows) | | |
|---|---|---|---|---|---|---|---|
| | | NRMSE ↓ | PPD ↓ | SSDD | NRMSE ↓ | PPD ↓ | SSDD |
| NPE | - | 0.312 (6.8e-03) | 0.117 (4.2e-03) | 0.463 (9.2e-03) | 0.315 (7.3e-03) | 0.112 (4.0e-03) | 0.386 (4.0e-02) |
| NNPE | - | **0.154 (6.1e-04)** | **0.035 (2.4e-04)** | 0.019 (1.7e-03) | 0.214 (7.4e-03) | 0.064 (4.2e-03) | 0.071 (1.5e-02) |
| NPE-DANN | 0.01 | 0.217 (3.2e-03) | 0.065 (8.7e-04) | 0.017 (1.8e-03) | 0.180 (7.9e-04) | 0.056 (1.6e-03) | 0.016 (1.2e-03) |
| NPE-DANN | 0.10 | 0.211 (4.4e-03) | 0.057 (2.2e-03) | 0.015 (9.6e-04) | **0.178 (1.7e-02)** | **0.045 (5.1e-03)** | 0.014 (6.1e-04) |
| NPE-DANN | 1.00 | 0.240 (9.0e-03) | 0.068 (3.6e-03) | 0.015 (4.3e-04) | 0.201 (2.0e-02) | 0.051 (8.0e-03) | 0.014 (2.4e-04) |
| NPE-MMD | 0.01 | 0.268 (4.2e-03) | 0.085 (3.1e-03) | 0.016 (6.4e-04) | 0.262 (1.1e-03) | 0.085 (6.4e-04) | 0.016 (3.9e-04) |
| NPE-MMD | 0.10 | 0.245 (6.3e-03) | 0.071 (3.6e-03) | 0.013 (1.5e-04) | 0.181 (9.7e-03) | 0.047 (3.8e-03) | 0.012 (3.0e-04) |
| NPE-MMD | 1.00 | 0.321 (4.2e-04) | 0.129 (3.4e-04) | 0.000 (3.1e-06) | 0.322 (5.3e-04) | 0.129 (2.9e-04) | -0.000 (2.7e-06) |

Table B.4: **Experiment 3**. Overview of the metrics in the different misspecification scenarios (mean and standard deviation of three runs). Please refer to Table 2 for a detailed description. Note that each standard deviation is given for a constant set of hyperparameters, so it only covers the computational uncertainty for a given setting. As shown by the performance changes when changing $\lambda$, hyperparameters have a large influence on the results, and different hyperparameter choices might lead to qualitative changes in the results.

### B.6 Experiment 4 - Decision Making

**Real-World Data**  We utilize empirical response time data from Implicit Association Test (IAT) experiments conducted in online settings and provided by Project Implicit (Xu et al., 2014). In the IAT, participants categorize words and images into two target categories as quickly and accurately as possible, aiming to respond swiftly while minimizing errors. We use a subsample from the standard Race IAT data set, collected between late 2014 and early 2015, with an initial sample size of $N = 166,283$. For each person, we use 120 experimental trials, 60 from each of the main experimental conditions in the IAT.

**Probabilistic Model**  As our cognitive model, we employ the diffusion decision model (DDM; Ratcliff et al., 2016). The DDM models the observed choices and reaction times as outcomes of a noisy evidence accumulation process and has been successfully applied to this and similar data sets (Klauer et al., 2007; Kvam et al., 2024). We base our modeling approach on previous work with the same data set (von Krause et al., 2022; von Krause & Radev, 2025, see these papers for the exact model formulation). To capture differences between experimental conditions—typically labeled congruent and incongruent—we specify separate drift rates and boundary separation parameters for each condition. Additionally, we introduce distinct non-decision time parameters for correct and error responses, to account for how error response times are recorded in the Project Implicit data set (i.e., the error response time is not saved directly, but only after the initially erroneous answer has been corrected). We adopt broad yet informative priors based on previous modeling studies utilizing the same data (von Krause & Radev, 2025), to maximize the closeness of our approach to real-world applications.

**Network Architecture**  For the summary network $\phi$, we utilize a deep set architecture (Zaheer et al., 2017) that reduces the input (response time, accuracy, and experimental condition of 120 trials per person) to a 12-dimensional representation. We use a dropout rate of 0.05 to avoid overfitting due to offline training. The generative inference network $q$ is based on an affine coupling flow model (Ardizzone et al., 2021; Kingma & Dhariwal, 2018), consisting of six invertible layers. As before, the domain classifier $\psi$ used in NPE-DANN is a conventional feedforward neural network with three hidden layers, each comprising 256 units. We omit both label smoothing and any reweighting of the gradient reversal component. We use an AdamW optimizer with an initial learning rate of $1 \cdot 10^{-4}$ and cosine decay.

**Training and Evaluation Details**  All neural networks are trained offline using a fixed simulation budget of $32,000$ data sets. Each model is trained for 100 epochs with a batch size of 32, resulting in $1,000$ iterations per epoch. For validation during training, we generate an additional $1,000$ simulated data sets. In the case of NPE-UDA methods, we also use $32,000$ empirical data sets for training and another $1,000$ for validation. We use $S = 100$ posterior samples per method and test data set to keep computation times reasonable.

To evaluate model performance, we construct four distinct test sets from the empirical data. We begin by categorizing the remaining $133,283$ examples—after removing training and validation data—into well-specified and misspecified subsets. Well-specified data sets closely resemble the simulated data, whereas misspecified ones are more likely to reflect model mismatch. To define this distinction, we use an out-of-distribution detection approach in summary space inspired by Schmitt et al. (2024): We train three standard NPE networks with equal settings and extract 12-dimensional summary embeddings for $10,000$ simulated test sets. We then compute the 90th percentile of Mahalanobis distances within the embedding space for each network. Averaging the resulting quantiles across the three runs yields a threshold, which we use to classify empirical data sets: Those with embeddings within the threshold are labeled well-specified, while those exceeding it are considered misspecified. We calculate the mean Mahalanobis distance for the embeddings of each empirical test data set—using the covariance matrix derived from simulated data—averaged across the three networks. Based on these distances, we categorize each data set accordingly. For subsequent analyses, we use as test data all 730 data sets labeled as misspecified, along with a separate random subsample of $10,000$ well-specified examples.

Both well-specified and misspecified test sets undergo a data cleaning procedure, resulting in two versions of each: a raw (uncleaned) and a cleaned data set. Cleaning involves removing response times below 200 ms or above 10 seconds, as well as intra-individual outlier trials. These outliers are identified as trials with

log-transformed response times falling outside 1.5 interquartile ranges (IQRs) from the 25% or 75% quantiles, calculated separately for each participant, condition, and trial correctness.

For the NNPE method, we again apply the spike-and-slab procedure to introduce noise into the input data—in this case, the response times. To prevent any sign reversal due to contamination, we enforce a lower bound by setting any negative response time values to 0.01 seconds.

**Metrics** As our first evaluation metric, we compute the MMD between the embedding spaces of simulated test data and uncleaned empirical test data. We hypothesized that UDA methods would yield better alignment—i.e., lower MMD values—compared to standard NPE, reflecting improved generalization to empirical data.

Our second metric is concerned with external validity. We examine correlations between individual participants' posterior median cognitive parameters and their age. Prior research has shown robust age-related effects, particularly on the boundary separation parameter and non-decision time (for correct trials). We therefore compute these correlations across all networks and test sets to evaluate the extent to which these known effects are recovered.

Our third metric focuses on posterior predictive distances. For each participant in the empirical test set, we use our trained networks to approximate 100 samples from the joint posterior distribution over cognitive model parameters. Using these samples, we re-simulate data and compare them to the original empirical data using root mean square error (RMSE) on a set of summary statistics: mean accuracy and response time quantiles (10%, 30%, 50%, 70%, and 90%), split by condition and further separated into correct and error trials. RMSEs are averaged across posterior samples, participants, quantiles, and experimental conditions. This yields one RMSE value each for accuracy, correct response times, and error response times per estimation method and per test data set (i.e., well-specified/clean, misspecified/clean, well-specified/unclean, and misspecified/unclean).

**Additional Results** Our last metric again uses MMD, but this time to quantify the discrepancy between the prior distributions of cognitive parameters and their corresponding posterior distributions after inference on empirical data. We utilize this divergence between prior and posterior distributions of cognitive parameters to test the hypothesis of Huang et al. (2023) that increasing $\lambda$ values lead to a convergence of the posterior to the prior. Figure B15 shows the results, with little evidence for a convergence to the prior up to $\lambda = 1.0$. However, we observe a sharp decrease in the distance between prior and posterior distributions for $\lambda = 10$ (not shown), suggesting that the model learns less from the data at this level of UDA influence. Similarly, while the lowest summary space distances are observed at $\lambda = 10$, the PPD reveals that, at this setting, the estimated parameters fail to adequately reproduce the empirical data.

Together, these findings highlight the importance of carefully balancing the UDA component in training: while stronger alignment can improve summary-level similarity, excessive emphasis may drastically impair parameter recovery and reduce the informativeness of the posterior. As parameter estimation effectively fails for the $\lambda = 10$ networks, we exclude these from further analyses.

Finally, we also compared results for shorter training time (50 instead of 100 epochs) and uniform instead of informative priors for the DDM parameters. In both cases, the alternative settings lead to worse performance across metrics and methods, so we omit these results for reasons of brevity.

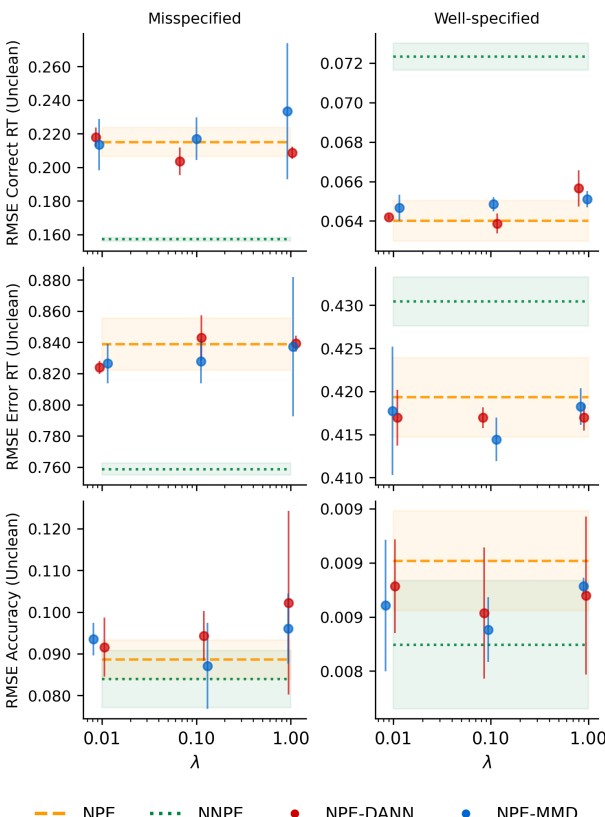

Figure B14: **Experiment 4**. Posterior predictive distance (RMSE) to **uncleaned** empirical response time data, averaged across three runs per method. Across run standard deviations are shown as shaded areas (for NPE and NNPE) or error bars (for NPE-DANN and NPE-MMD). The left column shows results for misspecified data sets, while the right column shows results for the (much larger) well-specified data set. The first row shows the network's prediction error for response times (in seconds) on correct trials, averaged across posterior samples, participants, response time quantiles, and experimental conditions. The second row shows the same metric for error response times, while the third row shows results for accuracy rates (in %). Please note that y-axis scales differ across subplots. $\lambda$ = UDA weight.

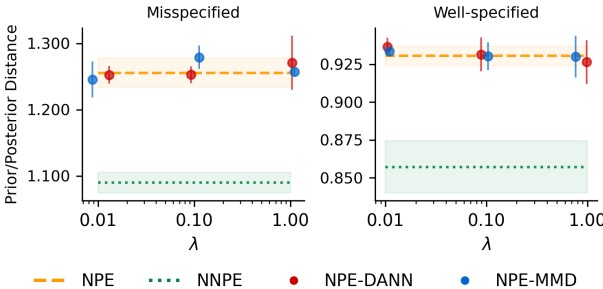

Figure B15: **Experiment 4**. MMD of prior vs. posterior distributions in the parameter space, averaged across three runs per method. Across run standard deviations are shown as shaded areas (for NPE and NNPE) or error bars (for NPE-DANN and NPE-MMD). The left column shows results for misspecified data sets, while the right column shows results for the (much larger) well-specified data set. Please note that y-axis scales differ across subplots. $\lambda$ = UDA weight.

