# OpenReview forum: "Does Unsupervised Domain Adaptation Improve the Robustness of Amortized Bayesian Inference? A Systematic Evaluation"
_TMLR — Accepted by TMLR_

### Review · Reviewer_XfNk · 2025-06-04

**Summary Of Contributions:**

The authors empirically assess approaches to ameliorating the effects of the distribution shift inherent in Amortized Bayesian Inference (ABI), where simulated data is used to train a (neural network) map from data to the posterior over parameters. They primarily consider 3 approaches: noisy neural posterior estimation (NNPE), NPE-MMD and NPE-DANN. NPE-MMD includes a loss term penalizing distribution shift between embeddings of the true and simulated data. DANN uses an adversial approach vs a classifier that tries to distinguish true vs real data embeddings. Their key finding is that UDA can help with a mis-specified likelihood (e.g. accounting for unmodeled noise processes) but can actually hurt performance under a mis-specified prior, since the weakened likelihood has less signal to counteract the bad prior. They also explore/discuss the utility of different metrics for assessment.

**Audience:**

Yes

**Claims And Evidence:**

Yes

**Requested Changes:**

I'd be interested to understand whether there is different behavior for prior mis-specification that is 1) too high variance vs 2) too low variance. Intuitively one would expect that using relatively broad/high variance/uninformative priors would be "safer", but presumably there is some trade-off.

**Strengths And Weaknesses:**

The paper is well written although 1) very heavy on acronyms (I'm not sure this is avoidable but a table of them might help) and 2) somewhat on the long side- there isn't content I would obviously suggest moving to an appendix but I would encourage the authors to do an editing sweep for brevity/clarity. Figure 1 is a nice exposition of NPE-MMD and NPE-DANN. Maybe I missed it but was NNPE not really described? A couple of sentences might be worth it to make the manuscript more self-contained.

I don't have any substantial technical criticisms of the manuscript. I would have been interested to see an application in population genetics since ABI is being used quite widely in that context. One could imagine training on a simpler demographic model (e.g., one that doesn't include migration) and asking if UDA improves performance estimating parameters like TMRCA for data simulated under a more complex model (e.g., including migration). However, given the length of the paper already this would probably be too much.

---

> ### Author Response · Authors · 2025-09-05
> **Response to Reviewer XfNk**
>
> We thank you for your thorough examination of our work and your constructive feedback. We are glad that you found our paper well written and technically sound. Below, we address your comments and questions.
>
> ## Writing
>
> Thank you for your positive feedback. We agree that the manuscript is quite dense with acronyms and performed an editing sweep, reducing redundant acronyms and including the great suggestion of an acronym overview table.
>
> ## NNPE
>
> The intuition behind the method is briefly described in the “Robust ABI” paragraph of the related work section and more details are provided in Appendix B1. In the revised version, we also make our rationale behind choosing NNPE as a baseline robust method for experimentation more clear (please see the "Robust methods in experimentation" section of our response to reviewer o89H). Do these explanations address your remark or is there any information we can add to make the explanation of NNPE more understandable?
>
> ## TMRCA
>
> Thanks for sharing this interesting application! While we agree that the paper is already quite dense, we will definitely keep it in mind for future research.
>
> ## Variance
>
> Thank you for raising this interesting point, which we agree is very relevant due to the prevalence of leaning towards broad priors in practice, as well as NNPE realizing the related approach of broadening the scope of simulation. We extended our empirical investigation of prior scale misspecification in Experiment 2, which tests the scenario of too narrow priors (i.e., test data generated with $\tau_0 = 3$ and $\tau_0 = 5$, while the training data was simulated with $\tau_0 = 1$), with the complementary scenario of too broad priors (i.e., $\tau_0 = 1/3$ and $\tau_0 = 1/5$): https://anonymous.4open.science/api/repo/JET-TMLR-EA67/file/under_vs_over_observed_data_gen_params_jet_weight_1.0.pdf?v=b8a7691d
>
> For easier comparison, we directly plotted the results as a function of $\tau_0$, so that the existing prior scale results from Figure 5 are displayed in the right half of the plots, while the new results are shown in the left half. We interpret these results as follows:
>
> - We indeed find that, for this experimental setting, too broad priors are “safer” in NPE than too narrow priors.
> - The symmetric summary space domain distance (SSDD) of NPE (partially overlapping with NNPE in the plot) when deviating from the well-specified setting suggests that both regimes are equally atypical from the perspective of the trained network. However, while we observe drastic differences between the analytic posterior and the NPE methods in terms of parameter recovery error (NRMSE), posterior predictive data space distance (PPD), and calibration (ECE) in the too-narrow-priors regime (right side), these errors are much less pronounced in the too-broad-priors regime (left side).
> - Therefore, the differences between the various NPE methods in the too-broad-priors regime are mainly constrained to different contraction (1 - PC) vs. calibration (ECE) trade-offs. Here, the contraction patterns mirror the general trend of NPE-UDA posteriors across our experiment to "ignore" unexpected information in the observed data and thus stay closer to the prior than NPE (i.e., less contraction equals broader NPE-UDA posteriors in the too-broad-priors regime and vice versa).
> - We hypothesize that the dimensionality of the inference problem plays a crucial role: In this low-dimensional (2D) inference task, the too-broad-priors regime is a much less clear misspecification/out-of-distribution setting than the too-narrow-priors regime, since in the former regime, the test setting is still well-represented in the typical set of the simulated training data. As dimensionality increases, we expect this to become less and less the case due to the curse of dimensionality. Thus, for high-dimensional inference problems, we expect the choice of overly broad priors to become as detrimental as overly narrow priors.

---

### Review · Reviewer_o89H · 2025-07-09

**Summary Of Contributions:**

The paper investigates unsupervised domain adaptation (UDA) applied to amortised simulation-based inference, specifically by making neural posterior estimation (NPE) more robust to model misspecification.
It contributes a novel robust approach for amortised Bayesian inference, NPE-DANN, that augments the standard NPE loss with a domain-adversarial classifier.
The paper also re-examines NPE-MMD, extending it to be amortised and to include alternative hyperparameter choices.
The authors explain and empirically verify that UDA implicitly learns a denoised posterior, which helps under likelihood misspecification, yet can worsen performance under prior misspecification.
Finally, experimentation was done on four examples that show NPE-DANN and NPE-MMD achieve good performance in well-specified and misspecified (likelihood) settings, supported by a thoughtful selection of evaluation metrics and clear justification. Open questions and obstacles to using NPE-DANN are addressed.

**Audience:**

Yes

**Claims And Evidence:**

Yes

**Requested Changes:**

Overall, this is an enjoyable and well-written paper. While some open questions and obstacles remain for the use of NPE-DANN, the authors acknowledge these limitations and address them well. The manuscript is interesting and well-executed, and in my view is appropriate for TMLR as is.

Some optional changes following weaknesses:
- Some more results or details on computational costs - while it makes sense that runtime increases minimally, it might still be good to see some more empirical results or justification.
- Regarding "a closer examination of UDA mechanisms is necessary to determine the nature of domain adaptation", while interesting future work given, is there anything people can do right now for this (e.g., posterior predictive checks)?
- Any remaining points in weaknesses not already clear in paper.

Spelling and grammar:
- p7. "intervalst"
- p8. Experiment 5.1 "uncontamined"

**Strengths And Weaknesses:**

Strengths:
- Clear and well-written.
- First adaptation to domain-adversarial neural networks to amortised NPE. NPE-DANN convincingly shown to be capable of decent performance for the goal of robust inference.
- Thoughtful updates to NPE-MMD. Nice to see useful updates (amortised, looked into hyperparameters) and comments on this, as well as connections to UDA.
- Experimentation well-done, interesting and insightful comments.
- Recognise and explicitly confront potential issues with prior misspecification.


Weaknesses:
- Hyperparameters and neural architecture choice. Both DANN and MMD introduce several extra hyper-parameters, increasing the tuning effort and computational cost. Although the paper notes this limitation, it still poses a hurdle to practical adoption. In addition, the learning summary approach requires custom summary networks and, for DANN, an accompanying discriminator tailored to each new domain, further expanding the set of design decisions the practitioner must manage.
- Adversarial stability - acknowledged in the paper, but training stability might pose an obstacle for practitioner use. I am unsure on any specifics for DANN, wonder if any mitigations/techniques were found to be needed in experimentation for stable training.
- Computational cost. Only briefly mentioned time per epoch in experiment 3. While it makes sense that NPE-UDA and NPE-MMD would have minimal impact on runtime, I wouldn't mind seeing a touch more concrete results on this across factors such as: training time, inference time, if we might expect more simulations to be required, if hyperparameter search is expected and would lead to increased overall computational cost.
- Model building. The paper does not really cover the case where the simulator is inadequate (fails to reproduce key features of observed data that the modeller cares about). It remains a bit unclear at which stage NPE-DANN (or a more general workflow) would flag such misspecification and what the modeller should do next (e.g., modify the simulator, redesign summary network, or abandon UDA).
- Unresolved handling of prior misspecification. The authors clearly show that UDA can break down when the assumed prior is wrong. Perhaps more guidance could be given on how users should remedy such a mismatch. Perhaps out of scope for paper, but if the authors have any useful comments on what could be done (modeller should consult more general Bayesian literature on prior misspecification, not use UDA, switch (or combine) with alternative robust SBI approaches?).
- Novelty. The methodology is not highly novel, given NPE-MMD (Huang, 2023) and NPE-UDA (Swierc, 2024). That said not an issue for TMLR, especially given the useful insights and updates from the extensive experimentation here.
- On robust methods in experimentation. I wonder why the NNPE from Ward's 2022 paper was used here, rather than the RNPE from the same paper? Also wonder if amortised cost estimation would be worth comparing against.

---

> ### Author Response · Authors · 2025-09-05
> **Response to Reviewer o89H (1/2)**
>
> We thank you for your in-depth review of our work. We are delighted that you found our paper to be enjoyable and well-written, and the experimentation insightful. We also appreciate your thoughtful and constructive feedback (thanks for spotting the typos!), and address your raised questions here and in the manuscript as detailed below.
>
> ## Hyperparameters
>
> We fully agree on this practical limitation of NPE-UDA approaches and believe that the large-scale Bayesian hyperparameter optimization in Experiment 1 (please see Figure B1 for further results), as well as the examination of different regularization parameter settings in all experiments, provides valuable insights on this issue. We revised our discussion to include not only the most sensitive regularization hyperparameter but also the broader issue of introducing additional hyperparameters.
>
> ## Adversarial stability
>
> We agree that this is an important hurdle for practical application. As described in the discussion, we found the setting of the regularization hyperparameter to have a decisive influence on the tendency of NPE-DANN to exhibit the typical instabilities inherent to adversarial training. Appendix B.1 details further training sensitivities that we found, with lower learning rates and longer training times stabilizing NPE-UDA methods generally. For NPE-DANN specifically, we tested the potentially stabilizing effect of label smoothing in Experiment 1, but the constant variability across different label smoothing settings in Figure B1 suggests that label smoothing does not stabilize adversarial training in NPE-DANN.
>
> ## Computational cost
>
> Thanks for raising these interesting points! While we agree about the importance of considering computational cost, we believe that the already quite dense manuscript profits from keeping this aspect brief due to the following reasons:
> - Time efficiency: NPE-UDA methods only change training time but not inference time since the modifications compared to standard NPE pertain exclusively to the training process. For concrete time measurements, we focus on the training times for Experiment 3 since it features the largest summary space and thus constitutes an upper bound measurement of the time increase due to integrating UDA methods with NPE in our experiment. In our revised version, we added these explanations and, to make the practical impact more tangible, that these measurements amount to a wall clock training time of ~ 4 minutes and 20 seconds for NPE-DANN instead of ~ 4 minutes for NPE in Experiment 3.
> - Simulation budget: While we always compare all methods for the same simulation budget as well as training epoch budget, the sensitivity of NPE-UDA methods to shorter training times (see Appendix B1) does indeed suggest longer training times to be beneficial in practice.
> - Hyperparameter search: We agree that this is expected in practice. However, as we argue in the discussion, we believe that this fundamental issue remains an open problem beyond the scope of this work due to the broader challenges of tuning UDA methods on label-free real-world data.
>
> ## Model building / inadequate simulator
>
> We agree that in the lighter misspecification scenarios, the simulator can still provide an adequate description of the data. However, the drastic decrease in performance in the scenarios with stronger misspecification, such as mu_0 = 5 or tau = 20 in Experiment 2, indicate that our experiments cover cases where the simulator is fundamentally inadequate for describing the observed data. We outline further approaches for flagging misspecification in the "Is there anything people can do right now for this (e.g., posterior predictive checks)?" section below.
>
> ## Handling of prior misspecification
>
> Our preferred approach coming from a Bayesian workflow perspective (Gelman et al., 2020) would be to detect prior misspecification via prior predictive / prior pushforward checks. We now explicitly address this in the discussion section, following our conclusion about the sensitivity of NPE-UDA methods to prior misspecification: "Consequently, prior predictive or pushforward checks (Bürkner et al., 2025; Box, 1980) become an even more essential part of the Bayesian workflow (Gelman et al., 2020) for NPE-UDA methods.". Crucially, prior elicitation remains one of the major challenges in Bayesian analysis (see Mikkola et al., 2024, and Bockting et al., 2025, for recent developments).
>
> ## Novelty
>
> Please refer to the “Contributions” part of our response to reviewer UKEB for our thoughts on this matter.

---

> > ### Author Response · Authors · 2025-09-05
> > **Response to Reviewer o89H (2/2)**
> >
> > ## Robust methods in experimentation
> >
> > Thanks for highlighting this interesting point! We acknowledge that NNPE was not the main focus of Ward et al. (2022), but rather RNPE. Thus, we see NNPE’s role in our benchmarks to be less about the specific noise augmentation scheme and more about "an instantiation of a simple additive noise training modification (Cranmer et al., 2020; Bernaerts et al., 2023)" (from our related work section). RNPE as well as amortized cost estimation are promising approaches, but not suited for ABI since their reliance on MCMC sampling renders them non-amortized (please also see the related work section). Therefore, we believe that noise augmentation – the presumably most straightforward current approach for robustifying ABI in practice –  is the best available amortized baseline method for our experiments. We revised our manuscript to communicate this rationale behind choosing NNPE more clearly.
> >
> > ## Is there anything people can do right now for this (e.g., posterior predictive checks)?
> >
> > We agree that it would be helpful to offer a more comprehensive discussion of this central practical problem, elaborating further approaches for measuring robustness besides the posterior predictive checks already discussed. We added the following elaboration to the "Measuring Robustness in the Real World" paragraph of the discussion:
> >
> > "Alternative approaches to measuring the robustness of NPE in the real world are also challenging to realize for NPE-UDA methods: The summary space is actively reshaped by NPE-UDA methods and therefore loses its validity as a proxy for detecting model misspecification (Schmitt et al., 2024). In contrast, the predictive variability of a deep ensemble
> > (Lakshminarayanan et al., 2017) on the observed data, which directly measures the predictive fragility resulting from model misspecification in NPE (Elsemüller et al., 2024), remains a meaningful diagnostic for NPE-UDA. However, using deep ensemble variability to guide hyperparameter search is problematic: Ensemble variability can straightforwardly be minimized by collapsing the posterior to the prior, thus rewarding excessive regularization during hyperparameter search. While such collapses could be partially addressed by jointly optimizing hyperparameter search for posterior contraction, doing so introduces the additional challenge of balancing these two search objectives in label-free real-world settings. Furthermore, embedding deep ensembles into hyperparameter searches multiplies the already large computational cost of the latter and consequently remains impractical."
> >
> >
> > ## References
> >
> > Yves Bernaerts, Michael Deistler, Pedro J Gonçalves, Jonas Beck, Marcel Stimberg, Federico Scala, Andreas S Tolias, Jakob Macke, Dmitry Kobak, and Philipp Berens. Combined statistical-mechanistic modeling links ion channel genes to physiology of cortical neuron types. bioRxiv, pp. 2023–03, 2023.
> >
> > Florence Bockting, Stefan T Radev, and Paul-Christian Bürkner. Expert-elicitation method for non-parametric joint priors using normalizing flows. Statistics and Computing, 35(5):132, 2025.
> >
> > George EP Box. Sampling and Bayes’ inference in scientific modelling and robustness. Journal of the Royal Statistical Society Series A: Statistics in Society, 143(4):383–404, 1980.
> >
> > Paul-Christian Bürkner, Marvin Schmitt, and Stefan T Radev. Simulations in statistical workflows. arXiv preprint arXiv:2503.24011, 2025.
> >
> > Kyle Cranmer, Johann Brehmer, and Gilles Louppe. The frontier of simulation-based inference. Proceedings of the National Academy of Sciences, 117(48):30055–30062, 2020.
> >
> > Lasse Elsemüller, Hans Olischläger, Marvin Schmitt, Paul-Christian Bürkner, Ullrich Köthe, and Stefan T Radev. Sensitivity-aware amortized Bayesian inference. Transactions on Machine Learning Research (TMLR), 2024.
> >
> > Andrew Gelman, Aki Vehtari, Daniel Simpson, Charles C Margossian, Bob Carpenter, Yuling Yao, Lauren Kennedy, Jonah Gabry, Paul-Christian Bürkner, and Martin Modrák. Bayesian workflow. arXiv preprint arXiv:2011.01808, 2020.
> >
> > Balaji Lakshminarayanan, Alexander Pritzel, and Charles Blundell. Simple and scalable predictive uncertainty estimation using deep ensembles. Advances in neural information processing systems, 30, 2017.
> >
> > Petrus Mikkola, Osvaldo A Martin, Suyog Chandramouli, Marcelo Hartmann, Oriol Abril Pla, Owen Thomas, Henri Pesonen, Jukka Corander, Aki Vehtari, Samuel Kaski, Paul-Christian Bürkner, and Arto Klami. Prior knowledge elicitation: The past, present, and future. Bayesian Analysis, 19(4):1129–1161, 2024.
> >
> > Marvin Schmitt, Paul-Christian Bürkner, and Ullrich Köthe. Detecting model misspecification in amortized Bayesian inference with neural networks. Proceedings of the German Conference on Pattern Recognition (GCPR), 2024.
> >
> > Daniel Ward, Patrick Cannon, Mark Beaumont, Matteo Fasiolo, and Sebastian Schmon. Robust neural posterior estimation and statistical model criticism. Advances in Neural Information Processing Systems, 35:33845–33859, 2022.

---

### Review · Reviewer_UKEB · 2025-08-01

**Summary Of Contributions:**

The paper identifies a key shortcoming in existing applications of amortized Bayesian inference (ABI) under model misspecification and introduces a regularization-based approach to enhance robustness. The proposed model structure is simple yet effective in addressing this issue. The evaluation is thorough, covering both synthetic and real-world datasets, and highlights the trade-offs and practical limitations of applying unsupervised domain adaptation (UDA) to ABI.

**Audience:**

Yes

**Claims And Evidence:**

Yes

**Requested Changes:**

It would strengthen the paper if the authors could address some of the weaknesses outlined above. In particular, demonstrating a more substantial methodological or theoretical advancement—beyond combining existing ABI and UDA techniques—would significantly improve the paper’s contribution. Without such progress, the innovation may be seen as limited.

**Strengths And Weaknesses:**

$\textbf{Strengths}$

1.The paper introduces and systematically evaluates a novel amortized version of NPE-DANN, expanding the applicability of domain-adversarial training to amortized Bayesian inference for the first time.

2.The proposed framework is conceptually simple and easy to implement, making it practical for integration into existing ABI pipelines.

3.The empirical evaluation is comprehensive, covering a diverse range of synthetic and real-world scenarios, and provides valuable insights into when domain adaptation improves or harms inference.


$\textbf{Weaknesses}$


1.The identified fragility of ABI under model misspecification is well-known and not specific to ABI alone. Similarly, the application of domain adaptation is a generic remedy. The paper does not offer novel methodological contributions to either ABI or UDA individually; rather, it combines existing tools without clear innovation beyond integration.

2.The paper lacks theoretical contributions—for instance, it does not establish tighter generalization bounds or characterize how the proposed method narrows known robustness gaps.

3.The approach is highly sensitive to the choice of the regularization weight, and no principled method is provided for tuning it in real-world, label-free settings—limiting practical usability.

---

> ### Author Response · Authors · 2025-09-05
> **Response to Reviewer UKEB**
>
> We thank you for your thoughtful review of our paper. We appreciate your assessment of the comprehensiveness of our empirical evaluation and the value of our insights and findings, as well as your recommendations for further strengthening our work. Below, we address your comments.
>
> ## 1. + 2. Contributions
>
> We agree that the fragility of neural networks under distribution shifts is well-known. However, investigating this issue in the context of model misspecification in the emerging ABI field is a comparably recent development. Despite several methodological works highlighting this fundamental challenge (e.g., Cannon et al., 2022), checking for model misspecification is still not an established part of applied ABI workflows. Thus, we believe that an important contribution of our work lies in further raising awareness of the dangers of misspecification in ABI and gaining deeper insight into the behavior of ABI methods under various misspecification scenarios.
>
> Furthermore, we also agree that, in theory, domain adaption seems to be the straightforward solution for this sim2real problem in ABI. We believe that a central significance of our paper lies in thoroughly testing the emerging combination of ABI and UDA and *highlighting that this combination is not as straightforward as it theoretically seems*.
>
> We argue that this is a critical finding for ABI practitioners: They need to be able to trust their inferential results in the real world, where the lack of ground-truth renders validation challenging. Modelers cannot precisely know about the type(s) of misspecification they encounter in the real world, and previous research only investigating specific types of likelihood misspecification suggests that NPE-UDA methods straightforwardly increase robustness in the real world. We believe that uncovering critical failure modes in our systematic evaluation of a broad range of misspecification scenarios is a central contribution of our work, preventing biased inference in potential high-stakes applications. While UDA is an established field, ABI is a relatively new and rapidly evolving field, especially concerning its robustness under model misspecification. Since our work is one of the first to explicate and systematically explore the combination of UDA and ABI, we focus our conceptual contribution on the missing pieces for understanding the uncovered failure modes of this combination by differentiating between the targets of robustness in section 3.4 (“What Is the Target of Robustness?”).
>
> Thus, we believe the contributions of our work to be well aligned with TMLR’s emphasis on sound evidence and relevance for TMLR’s audience. We thank you for pointing out the lack of clarity in communicating our work’s contributions and revised our manuscript, especially the contributions section, to clearly communicate our reasoning above.
>
> ## 3. Regularization weight
>
> We also agree about the challenge that the sensitivity to the choice of the regularization weight poses for real-world applications. Previous research on NPE-MMD either suggested stability across a wide range of values for their particular non-amortized setting (Huang et al., 2023) or did not investigate the impact of this critical hyperparameter (Swierc et al., 2024). We therefore consider the sensitivity of NPE-UDA approaches uncovered in our systematic evaluation to be a further important contribution of our work. Additionally, we argue that, similar to the broader UDA field, there currently exists no simple solution for the challenging problem of tuning NPE-UDA methods in the real world.
>
> In the revised version, we extended the discussion section with a more complete discussion of potential methods for quantifying robustness in the context of ABI, such as posterior predictive measures (please see the "Is there anything people can do right now for this (e.g., posterior predictive checks)?" section of our response to reviewer o89H).
>
> ## References
>
> Patrick Cannon, Daniel Ward, and Sebastian M Schmon. Investigating the impact of model misspecification in neural simulation-based inference. arXiv preprint arXiv:2209.01845, 2022.
>
> Daolang Huang, Ayush Bharti, Amauri Souza, Luigi Acerbi, and Samuel Kaski. Learning robust statistics for simulation-based inference under model misspecification. Advances in Neural Information Processing Systems, 36, 2023.
>
> Paxson Swierc, Marcos Tamargo-Arizmendi, Aleksandra Ćiprijanović, and Brian D Nord. Domain-adaptive neural posterior estimation for strong gravitational lens analysis. arXiv preprint arXiv:2410.16347, 2024.

---

### Decision · Action_Editor_2owZ · 2025-09-21

**Recommendation:** Accept as is

**Audience:**

Yes

**Audience Explanation:**

This would be particularly interesting for any ML practitioner who trains an ML model using synthetic data that may not fully represent the real-life test data. It provides interesting results on how to align the the model for better test performance.

**Claims And Evidence:**

Yes

**Claims Explanation:**

This paper examines how machine learning models perform when trained with synthetic data, which may underperform or fail when applied to real-world scenarios where the test data differs significantly from the training data. The authors focus on unsupervised domain adaptation to align the test data with the in silico training data.

The reviewers were generally positive about the paper, and most of their suggestions were addressed. Reviewer UKEB suggested rejecting the paper because it did not present sufficient novelty. However, he recognises the educational and reproducibility benefits that this paper brings to the community and understands that the guidelines for TMLR differ from those in our field's conferences. The other two reviewers voted for acceptance.